# Ultrafast silicon photonic reservoir computing engine delivering over 200 TOPS

Dongliang Wang, Yikun Nie, Gaolei Hu ⓘ, Hon Ki Tsang ⓘ & Chaoran Huang ⓘ ✉

Reservoir computing (RC) is a powerful machine learning algorithm for information processing. Despite numerous optical implementations, its speed and scalability remain limited by the need to establish recurrent connections and achieve efficient optical nonlinearities. This work proposes a streamlined photonic RC design based on a new paradigm, called next-generation RC, which overcomes these limitations. Our design leads to a compact silicon photonic computing engine with an experimentally demonstrated processing speed of over 60 GHz. Experimental results demonstrate state-of-the-art performance in prediction, emulation, and classification tasks across various machine learning applications. Compared to traditional RC systems, our silicon photonic RC engine offers several key advantages, including no speed limitations, a compact footprint, and a high tolerance to fabrication errors. This work lays the foundation for ultrafast on-chip photonic RC, representing significant progress toward developing next-generation high-speed photonic computing and signal processing.

Artificial intelligence (AI) is one of the most transformative technologies of the 21st century. The rapid expansion of data volume has facilitated the development of increasingly large and complex neural network models, such as ChatGPT, Gemini, and Sora. However, this growth poses significant challenges for conventional electronic computing hardware, particularly in terms of speed and power consumption[1,2]. These limitations have become a major bottleneck in AI advancement, driving the exploration of innovative computing hardware. Photonics has emerged as a promising approach for neural network implementation due to its potential to drastically accelerate processing speeds while reducing power consumption and latency[3–16].

Photonic reservoir computing (RC) is gaining considerable research interest among various photonic neural network (PNN) architectures due to its ability to eliminate the need for meticulous training and configuration of every optical device to achieve optimal weights[17–28]. Derived from recurrent artificial neural networks, RC is a three-layer NN consisting of an input layer, a reservoir layer, and a readout layer, as shown in Fig. 1a[29]. The input layer receives information and performs initial processing, while the reservoir layer typically consists of nonlinear nodes with random, fixed connections; and the readout layer then combines signals from the reservoir layer to generate the desired output through training. Compared to conventional

PNNs, photonic RC offers several intrinsic advantages. In conventional PNNs, precisely tuning the optical connections between neurons is a significant challenge due to manufacturing variations and device crosstalk[10]. Moreover, training PNNs is more challenging than training their electronic counterparts, as determining the gradient of an optical system is complicated[30]. In contrast, photonic RC eliminates the need to train the optical reservoir layer; only the readout layer requires training. This simpler training process significantly reduces both training time and power consumption, making photonic RC a more efficient and practical approach.

In traditional photonic RC systems, realizing efficient optical nonlinearities (i.e., nonlinear nodes) is a fundamental challenge. To mitigate this problem, a widely adopted solution is to employ time-division multiplexing, wherein a single nonlinear node with time-delayed feedback is used to generate multiple virtual nodes in the time domain[21,31–33]. While this method is relatively easy to implement, it slows down processing speed due to time multiplexing, leading to a compromise between the number of nonlinear nodes and processing speed. Furthermore, the limited bandwidth of optoelectronic nonlinear components typically used in photonic RCs further restricts processing speed[21,27,28]. An alternative approach to overcome these speed limitations is to use spatially distributed nonlinear nodes[20,26,34].

Department of Electronic Engineering, The Chinese University of Hong Kong, Shatin, Hong Kong SAR, China. ✉e-mail: crhuang@ee.cuhk.edu.hk

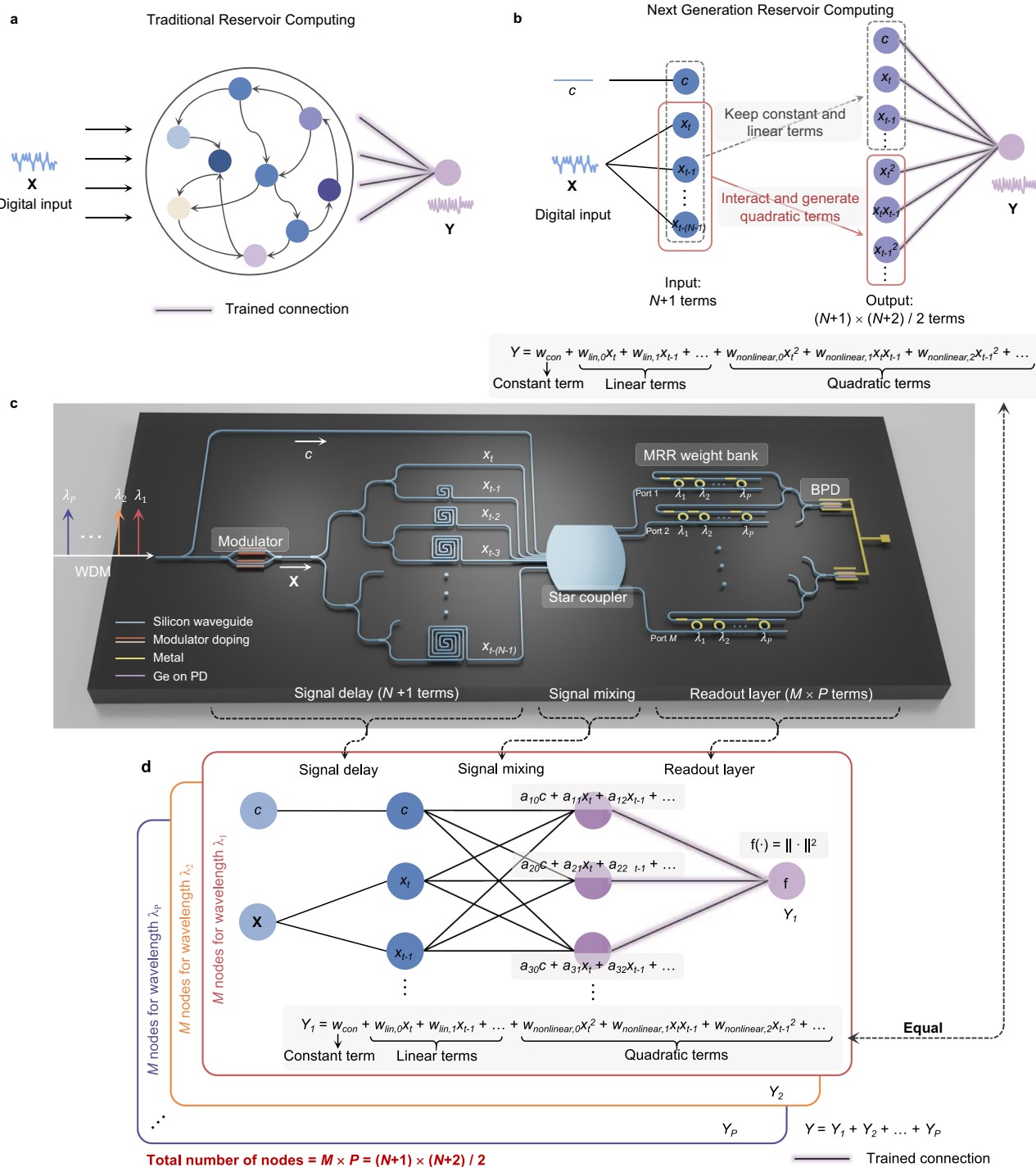

**Fig. 1 | Conceptual diagram of traditional RC, digital NG-RC, and photonic NG-RC engine. a** The framework of a traditional reservoir computing system, including an input layer, a reservoir layer, and a readout layer. **b** The principle of next-generation reservoir computing. The reservoir layer's input contains $N+1$ terms, and the reservoir layer's output contains $\frac{(N+1)(N+2)}{2}$ terms, which are the constant term, the linear terms, and the quadratic terms. **c** The scheme of our photonic NG-RC chip. The same input data **X** is encoded onto $P$ wavelengths of light. The input **X** is split into $N$ delayed copies sent into a star coupler together with an unmodulated laser representing the constant $c$. Input signals evolve within the star coupler with $M$ output ports. The readout layer is implemented optically using microring (MRR) weight banks and a balanced photodiode (BPD). **d** Mathematical principle of photonic NG-RC engine.

However, the scalability of such systems, particularly in terms of the number of nonlinear nodes, is hindered by the complexity of establishing recurrent interconnections between the nodes. This results in a trade-off between the number of photonic neuron nodes and the overall system footprint. As a result, achieving ultrafast and compact photonic RCs remains a significant challenge.

To address these challenges, we propose and experimentally demonstrate a novel integrated photonic RC chip based on a new framework called next-generation RC (NG-RC)[35]. NG-RC eliminates the need for recurrent connections by replacing the recurrent layer with a feedforward network driven by time-delayed inputs. This simplification not only streamlines the system architecture but also matches,

and in some cases surpasses, the computational performance of traditional RC systems. Building on the NG-RC framework, we design a high-speed, ultra-compact photonic RC system integrated on a silicon chip. Our RC chip employs a passive star coupler with delay-line waveguides to implement the feedforward network, effectively replacing the reservoir layer in conventional photonic RC systems that typically rely on complex components. This design enables integrating the reservoir layer on a silicon photonic chip with a footprint of just 2 mm², while still achieving best-in-class computational performance across various benchmarks and practical tasks, compared to much bulkier photonic systems[20–28,36,37]. Since our chip is built using linear optical devices, it overcomes the speed and bandwidth limitations typically seen in photonic RC systems. As a result, we achieve the fastest operation speed to date, with experimentally demonstrated information processing rates exceeding 60 GHz. This speed can be further enhanced to over 100 GHz with the use of higher-speed optical modulators and photodetectors[38,39].

In addition, our design features high scalability, energy efficiency, and remarkable tolerance to fabrication errors, making it well-suited for large-scale computing systems. The system is capable of supporting over 5000 output nodes, where the number of nodes directly correlates with computational capacity. We further demonstrate that this capacity can be further enhanced through wavelength-division multiplexing. Notably, NG-RC, with the same number of nonlinear nodes, typically outperforms many traditional RC systems in terms of computational performance. Moreover, the required chip area per nonlinear node and energy consumption in our design are far superior to those of other photonic and electronic systems, including state-of-the-art platforms like the GPU H100. This combination of high performance, high processing speed, compact design, and energy efficiency positions our integrated photonic RC chip as a strong candidate for next-generation computing.

## Results
### Principle
The core idea behind physical RC is designing and utilizing a dynamic system as a reservoir to map input data into higher dimensions[40]. Recent researches reveal that NG-RC can surpass conventional RC by requiring far fewer metaparameters to optimize, shorter training datasets, and significantly fewer node dimensions compared to comparable reservoir computers[41]. The principle of the NC-RG is shown in Fig. 1b. The NG-RC functions comparable to conventional RC, but it does not require building a complex recurrent NN with nonlinear nodes. Instead, it operates as a feedforward network, where the nodes represent different forms of transformations of the input (referred to as a feature vector). The output is then a linear regression of these elements in the feature vector (see Fig. 1b). Mathematically, such a feature vector can be expressed as:

$$\mathbf{O}_{total} = c \oplus \mathbf{O}_{linear} \oplus \mathbf{O}_{nonlinear} \tag{1}$$

where $\oplus$ represents the vector concatenation operation, $c$ is a constant, $\mathbf{O}_{linear}$ is the linear vector $\mathbf{X} = [x_t, x_{t-1}, ..., x_{t-(N-1)}]^T$, and $\mathbf{O}_{nonlinear}$ is the nonlinear transformation of the inputs. While it is flexible to select the nonlinear transformations, a simple quadratic polynomial of the inputs has been demonstrated to have good computing capabilities[35]. Therefore the nonlinear feature $\mathbf{O}_{nonlinear}$ is given by:

$$\mathbf{O}_{nonlinear} = \mathbf{O}_{linear} \otimes \mathbf{O}_{linear} \tag{2}$$

where $\otimes$ represents the outer product operation. It is worth noting that in the NG-RC, the dimension of the output vector $\mathbf{O}_{total}$ is significantly smaller than that in comparable RC systems[35]. When the input vector contains a signal of the length of $N$ and a constant, the size of the $\mathbf{O}_{total}$ only needs to be $\frac{(N+1)(N+2)}{2}$, which includes a constant $c$, $N$

inputs, and $\frac{N(N+1)}{2}$ quadratic polynomials of inputs. The result $y_{out}$ is the linear combinations of the elements in $\mathbf{O}_{total}$ with weights trained by linear regression.

Single wavelength operation: Fig. 1c, d illustrate our proposed photonic RC chip guided by the NG-RC framework. The input data $\mathbf{X}$ is encoded onto the amplitude of a laser with a wavelength of $\lambda$. The input is then split into $N$ delayed copies sent into a star coupler together with an unmodulated laser representing the constant $c$. The neighboring delay line introduces a time delay of $\Delta t$, which is equal to one symbol duration of the input signal. These delay lines ensure that input data at different times simultaneously reach the followed star coupler. After the star coupler, the output vector $\mathbf{y}_{star,\lambda}$ is expressed as:

$$\mathbf{y}_{star,\lambda} = \boldsymbol{\omega}_{star,\lambda} \cdot (c \oplus \mathbf{X}) \tag{3}$$

Here, $\boldsymbol{\omega}_{star,\lambda}$ denotes an $M \times (N+1)$ complex matrix, representing the transfer function of the star coupler at wavelength $\lambda$, where $N$ is the number of delayed input copies and $M$ is the number of outputs of the star coupler.

The outputs from the star coupler are followed by a readout layer, which can be implemented either digitally or optically. In a digital implementation, each output of the star coupler is detected by a photodetector, producing an output given by:

$$
\begin{aligned}
y_{PD,i} &= \left| \boldsymbol{\omega}_{star,\lambda,i} \cdot (c \oplus \mathbf{X}) \right|^2 \\
&= \underbrace{c_i}_{constant} + \underbrace{\sum_{n=1}^{N} a_{n,i} x_{t+1-n}}_{linear\ terms} + \underbrace{\sum_{m=1}^{N} \sum_{n=m}^{N} b_{mn,i} x_{t+1-m} x_{t+1-n}}_{quadratic\ terms}
\end{aligned} \tag{4}
$$

where $y_{PD,i}$ is the output of the photodetector and $c_i$, $a_{n,i}$, and $b_{mn,i}$ are constants, which are determined by the transfer function of the star coupler. Eq. (4) shows that the output of each photodetector is a combination of constant, linear inputs, and their quadratic polynomials. $M$ outputs form a vector $\mathbf{y}_{PD}$, where each element is a similar mixture but with different coefficients determined by the transfer function matrix of the star coupler. When the number of outputs $M \geq \frac{(N+1)(N+2)}{2}$, $\mathbf{y}_{PD}$ can be transformed into the feature vector required by the NG-RC, consisting of constant, linear, and nonlinear components, using a linear matrix. In this case, the system functions equivalently to the NG-RC. It is worth noting that the actual values in the transfer function of the star coupler do not affect the final computing result because the readout layer after the star coupler can be trained and adjusted to ensure optimal values. This feature makes our system have a high tolerance for fabrication errors.

To realize an optical readout layer, one approach is to pass the outputs from the star coupler through an array of programmable microring resonators (MRRs)(the first column of MRRs in Fig. 1c). The resonance wavelengths of the MRRs are aligned with the input wavelength. By adjusting the resonance of each MRR using an embedded thermal phase shifter, the fraction of light directed to the Drop ports can be finely controlled. In addition to tuning the signal's amplitude via the phase shifter on the MRR, a second phase shifter on the bus waveguide adjusts the signal phase. The combined tuning of these two phase shifters enables precise control of complex weights. The weighted signals are then detected by a balanced photodetector (BPD). The outputs from the first $\frac{M}{2}$ rows of MRRs are combined and detected by one photodiode, while the remaining MRR outputs are combined and detected by a second photodiode. The differential signal between the two photodiodes provides the full range of weights, including positive and negative values. The final signal after the BPD is

given by:

$$
\begin{aligned}
y_{BPD,\lambda} &= \left|\boldsymbol{\omega}^+_{MRR}\mathbf{y}^+_{star,\lambda}\right|^2 - \left|\boldsymbol{\omega}^-_{MRR}\mathbf{y}^-_{star,\lambda}\right|^2 \\
&= \underbrace{c}_{\text{constant}} + \underbrace{\sum_{n=1}^{N}\omega_{\text{lin},n}x_{t+1-n}}_{\text{linear terms}} + \underbrace{\sum_{m=1}^{N}\sum_{n=m}^{N}\omega_{\text{nonlinear},mn}x_{t+1-m}x_{t+1-n}}_{\text{quadratic terms}}
\end{aligned}
$$

(5)

where $\boldsymbol{\omega}^+_{MRR}$ and $\boldsymbol{\omega}^-_{MRR}$ are complex weight vectors given by the first and second half rows of MRRs, respectively. $\mathbf{y}^+_{star,\lambda}$ and $\mathbf{y}^-_{star,\lambda}$ are the output vectors from the first $\frac{M}{2}$ outputs and second $\frac{M}{2}$ outputs of the star coupler, respectively, given by Eq. (3). Eq. (5) indicates that the output of the BPD is a regression of the constant, inputs, and quadratic polynomials of inputs. The regression coefficients $c$, $\omega_{\text{lin}}$, and $\omega_{\text{nonlinear}}$ can be programmed from -1 to 1 using the MRR array. Therefore, the outputs of the optical engine have an equivalent function to those of the NG-RC. The derivation details of Eq. (5) are provided in the Supplementary Note 3.

For a given input port size $N+1$ driven by $N$ delayed input copies and one constant, the computing performance improves as the number of outputs $M$ of the star coupler increases, and this performance reaches saturation when $M = \frac{(N+1)(N+2)}{2}$. Increasing the computing performance requires a very slight increase in the size of the star coupler to accommodate more input and output waveguides, without changing its design. We estimate that a star coupler with an area of 6.25 mm² can support more than 5000 outputs, and this number is limited by the optical power to ensure a reasonable signal-to-noise ratio (SNR) (See analysis details in the Discussion and Supplementary Note 7).

Scaling to multi-wavelength operation: In addition to increasing the number of outputs $M$, the computing capability can be further scaled up through wavelength-division multiplexing (WDM). As shown in Fig. 1c, d, the input data **X** is encoded onto an array of WDM lasers with $P$ distinct wavelengths. At the output of the star coupler, the signals at different wavelengths are individually weighted by a column of MRRs. Each column is designed with a distinct resonance wavelength, aligned with the corresponding WDM laser, allowing precise control and weighting of the optical signals across multiple wavelengths. After the BPD, the output is the summation of the photoelectric currents generated by different wavelengths, which is given by $y_{BPD} = \sum_{i=1}^{P} y_{BPD,\lambda i}$. Employing WDM further increases the dimension of the output vector from $M$ to $M \times P$.

## Device design and experimental setup

Figure 2a shows the photonic chip used in our experiment, built on a silicon-on-insulator (SOI) platform. According to the general theory of universal approximations, the input size is ideally considered to be infinitely large[42]. However, empirical observation suggests that the Volterra series converges quickly, even with a relatively small input dimension[35]. Consequently, the number of delay lines does not need to be large. In our demonstration, our device uses 8 delay lines with incremental lengths to produce 8 delayed copies of the input signal, resulting in a compact photonic RC system. The signal is evenly divided into eight copies by on-chip splitters. Each copy experiences a different delay caused by an array of delay lines with incremental delay. The incremental delay is set at 16.7 ps (corresponding to 1.18 mm of silicon waveguide) to accommodate a 60 GBaud input signal. The star coupler occupies a footprint of 0.04 mm², with 9 inputs (one for unmodulated light and eight for delayed signal copies) and 45 (i.e., (9 + 1) × 9/2) outputs to generate the necessary feature vector. As discussed in the Principle section, mathematically, any full-rank transmission matrix from the star coupler would suffice. However, considering a lower power density at the edges compared to the center, we design the widths of the output ports to decrease quadratically from the edge to the center, as shown in Fig. 2c. This optimized design ensures that each output maintains a nearly uniform power distribution and SNR across all ports, as shown in Fig. 2d. The light transmission within the star coupler is simulated using Finite-Difference Time-Domain (FDTD) and shown in Fig. 2b. The star coupler together with delay lines functions equivalently to the input and reservoir layer in conventional RC systems but with a footprint of only 2 mm², significantly smaller than the reservoir layer in other RC systems. Method provides chip design and fabrication details.

The experimental setup is illustrated in Fig. 2a. We first demonstrate the operation of our NG-RC chip using a single wavelength, and

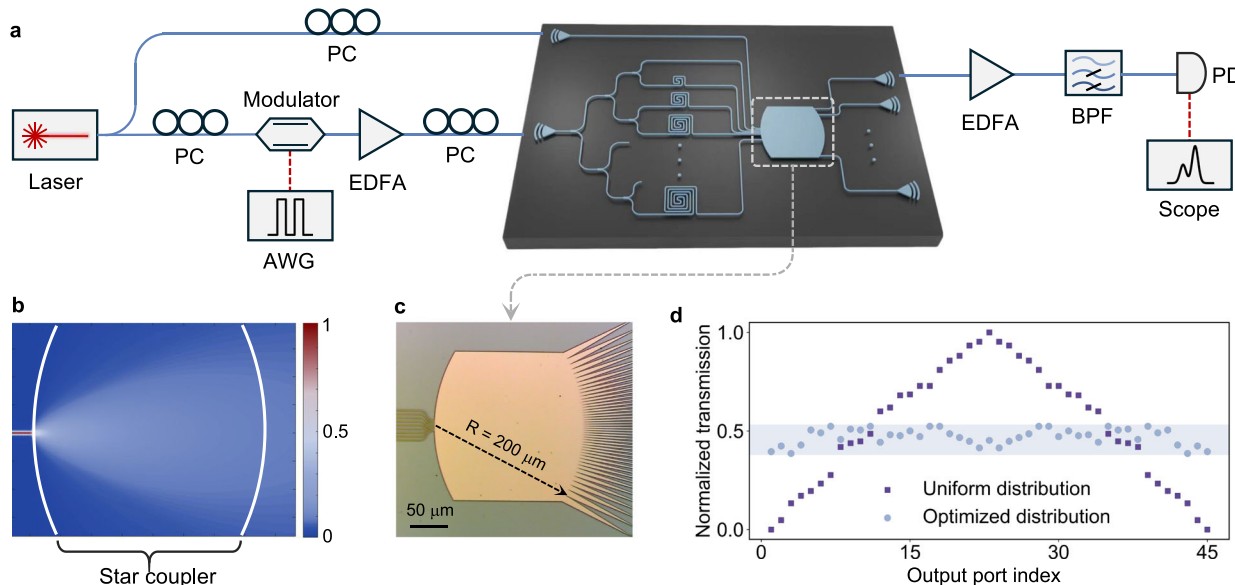

**Fig. 2 | Device design and experimental setup. a** Experimental setup with the fabricated device. PC, polarization controller. AWG, arbitrary waveform generator. EDFA, erbium-doped fiber amplifier. BPF, bandpass filter. PD, photodiode. **b** FDTD simulation result of light transmission in the star coupler. **c** Microscope image of the optimized star coupler. The diameter of the star coupler is 200 $\mu m$. **d** The normalized transmission of each output port when the width distribution of output ports is uniform or optimized.

extend the demonstration to multiple wavelengths in the later section. For the single-wavelength demonstration, we use a continuous-wave laser operating at 1550 nm as our light source. The light is split into two equal branches by a 50/50 coupler. One branch is coupled to the photonic chip as a constant reference $c$, while the other is modulated by a thin-film lithium niobate intensity modulator (LIOBATE, LB4C6PSBM63, 40 GHz bandwidth, RF $V_\pi \approx 3$ V). The input signal, generated by an arbitrary waveform generator (AWG, KEYSIGHT, M8199A, 256 GSa/s), is encoded onto the amplitude of the light by the modulator. To achieve a 60 Gbaud line rate, we set the sample rate of the AWG to 240 GSa/s. The modulator is configured in a push-pull arrangement and biased at the null point, corresponding to the minimum light intensity. The voltage applied to the modulator is set to $|V| < V_\pi/4$ ensuring that the input data can be linearly modulated onto the laser's amplitude. The modulated light is then amplified by an erbium-doped fiber amplifier (EDFA) before being injected into the photonic chip. In our experiment, the readout layer of the NG-RC is implemented digitally. The optical outputs are coupled out and detected sequentially using an off-the-shelf photodetector (COHERENT, model XPDV3120R-VM-FA, with a 70 GHz bandwidth) and digitized by a real-time oscilloscope (KEYSIGHT, model UXR0592AP, with a 59 GHz bandwidth and 256 GSa/s sampling rate). Finally, the readout layer is trained using a digital computer. As discussed in the Principle section, the readout layer can also be implemented optically and in real-time.

## Tasks for sequence prediction

We first evaluate the performance of our photonic NG-RC chip on sequence prediction through two benchmark tasks: the Santa Fe Laser dataset and the Lorenz63 system. In a prediction task, the goal is to predict the future output of a dynamical system based on the past output[43]. Prediction tasks can be classified into two categories: (i) when the system dynamics are unknown, and (ii) when the system dynamics are known.

The Santa Fe Laser dataset is generated from a chaotic system with unknown dynamics, consisting of experimental measurements of the optical power emitted by a laser operating in a chaotic regime[44]. In this task, the target is to train the photonic NG-RC chip to predict the next sampling point $x_{t+1}$ generated from the laser based on the previous sampling points, as shown in Fig. 3a. We use 6000 sampling points to train the readout layer, aiming to find the optimal set of 45 weights at the readout layer that satisfy $x_{t+1} = \boldsymbol{w}_{out}\boldsymbol{y}_{PD}$ using Tikhonov regularization. 2500 points are used for testing. Figure 3b, c show the predicted sequences from our photonic NG-RC system during the training and testing phases, both of which closely align with the target sequences. The prediction accuracy is assessed using the Normalized Mean Square Error (NMSE), defined as $NMSE = \frac{\sum_n [x'(n) - x(n)]^2}{\sum_n \left\{ x(n) - \frac{\sum_n [x(n)]}{n} \right\}^2}$, where $x'(n)$ and $x(n)$ represent the predicted and target sequence, respectively. The NMSE generated by our system is 0.029. In Fig. 3d, we compare the performance of our system with previously reported work[25,26,45–50] by showing their readout layer dimensions and the achieved NMSE. The results show that our photonic NG-RC system achieves the lowest NMSE with the smallest output dimension, while also demonstrating the fastest processing speed compared to the other systems.

We then demonstrate the effectiveness of our NG-RC chip using a Lorenz63 time series forecasting task. Lorenz63 dataset is generated from a chaotic system where the system dynamics are known and can be described by a set of three coupled ordinary differential equations[51]:

$$\dot{x} = 10(y - x), \dot{y} = x(28 - z) - y, \dot{z} = xy - 8z/3 \quad (6)$$

The Lorenz63 system serves as a paradigmatic example of deterministic chaos. In this task, the goal is to predict the next value of $z$ one step ahead using the current values of $x$ and $y$. This cross-prediction task is different and more challenging compared to most photonic reservoir computing demonstrations, which typically predict future $x$, $y$, $z$ values from past $x$, $y$, $z$ values[52,53]. This task holds practical significance, as it showcases the potential to use observable sensory data to infer unobservable information.

Figure 3e shows the workflow of our photonic NG-RC system for handling the Lorenz63 task. To generate the training and testing dataset, we solve Eq. (6) using a time step of 0.05. We use 400 data points for each variable $x$, $y$, $z$ for training and 600 data points for each variable for testing. The detailed data processing method for the Lorenz63 task is provided in Method. Figure 3f, g show the predicted $z$ sequence during the training and testing phase, which closely matches the ground truth. The correlation between the predicted and target $z$ values, based on 600 test data points, exceeds 0.99, as shown in Fig. 3h. The experimental NMSE for the Lorenz63 task is $1.43 \times 10^{-2}$ using only 45 output nodes. This NMSE can be further reduced by utilizing a modulator with a higher bandwidth (currently, a 40 GHz modulator generates a 60 Gbaud signal). In digital simulations, without bandwidth constraints, the NMSE can be reduced to $3 \times 10^{-4}$ with 45 nodes. In comparison, ref. 37 reports an NMSE of $0.9 \times 10^{-2}$ using over 2000 nodes.

## Tasks for system emulation

We further evaluate the ability of our photonic NG-RC chip for system approximation and emulation. In a system emulation task, the RC system is trained to approximate the response of a dynamical system[43]. Such approximations can be classified into two types, depending on whether the system dynamics are known or unknown. This task has practical applications in signal processing. Here, we demonstrate two tasks.

The first task is nonlinear channel equalization (NCE), which aims to recover the original signal from the noisy signal distorted by a nonlinear multipath RF channel. After successful training, our photonic NG-RC chip can be treated as an unknown nonlinear filter capable of removing the nonlinear noise in this system. The symbols of the original signal $d(n)$ are randomly chosen from the set $\{-3, -1, 1, 3\}$. The distorted signal $u(n)$ is defined as

$$
\begin{aligned}
q(n) = &\ 0.08d(n+2) - 0.12d(n+1) + d(n) + 0.18d(n-1) - 0.1d(n-2) \\
&+ 0.091d(n-3) - 0.05d(n-4) + 0.04d(n-5) + 0.03d(n-6) \\
&+ 0.01d(n-7)
\end{aligned}
$$

$$(7)$$

$$u(n) = q(n) + 0.026q(n)^2 - 0.011q(n)^3 \quad (8)$$

Figure 4a shows the workflow of emulating the perfect filter using our photonic NG-RC chip. In the NCE task, we use 3000 data points for training and 6000 for testing. The recovered waveforms from the training and testing phases are shown in Fig. 4b and c, respectively. During the training phase, the recovered signal closely matches the original signal. In the testing phase, minor errors are observed, resulting in a bit error rate (BER) of 0.06. This BER is measured at an experimental SNR of 12 dB. The relatively low SNR is due to the bandwidth limitations of the modulator. The experimental BER is lower than those reported in refs. 17,21,46,54 when the SNR is 12 dB. We further simulate the BERs when the SNR improves as shown in Fig. 4d. If a higher bandwidth modulator is used, the BER can be reduced to below $10^{-4}$.

The second system emulation task we perform is the Nonlinear Auto-Regressive Moving Average (NARMA10) task, which is one of the most-used benchmark tasks in the field of reservoir computing. The

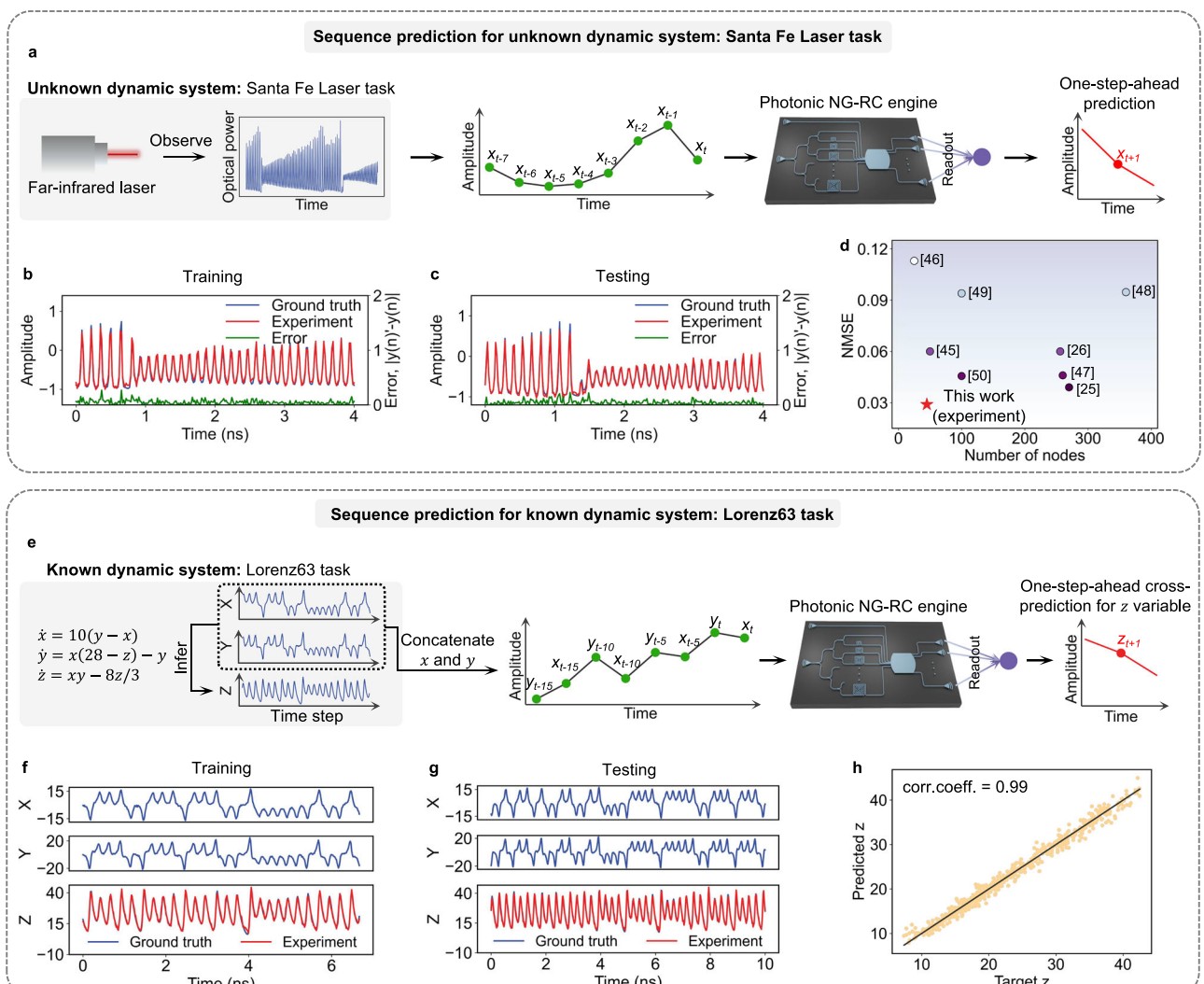

**Fig. 3 | Experimental demonstration of sequence prediction tasks. a** Workflow for performing Santa Fe Laser task using our photonic NG-RC chip. **b** The predicted sequence (red) and target sequence (blue) for the Santa Fe laser task during training. **c** The predicted sequence (red) and target sequence (blue) for the Santa Fe laser task during testing. **d** Performance comparison on the Santa Fe Laser task with other reported work, focusing on NMSE versus the number of nodes. **e** Workflow for predicting the *z* variable using the *x* and *y* variables from the Lorenz63 system.

The Lorenz63 is a known dynamic system. **f** Ground truth (blue) and prediction (red) variables of Lorenz63 during the training phase. *X*, and *Y* variables are fed into photonic NG-RC system, and the *Z* variable is predicted. **g** Ground truth (blue) and predicted (red) variables of Lorenz63 during the testing phase. **h** Correlation between predicted *z* and target *z*. The correlation coefficient, based on testing 600 data points, exceeds 0.99.

objective is to train our photonic NG-RC chip to replicate the behavior of a 10th-order nonlinear system, which has known dynamics modeled by the following recurrent equation:

$$y(n+1) = 0.3\mu(n) + 0.05y(n)\left[\sum_{i=0}^{9} y(n-i)\right] + 1.5\mu(n-9)\mu(n) + 0.1 \quad (9)$$

where $\mu(n)$ is the system input, sampled from a uniform distribution within [0, 0.5], and $y(n)$ is the system output. We train our system to generate the output $y(n+1)$ based on input $\mu(n)$. Figure 4e shows the workflow for emulating the NARMA10 dynamic system using our photonic NG-RC chip. We use 1000 sampling points for training and another 1000 sampling points for testing. Detailed data processing methods for the NARMA10 task are provided in Method. The experimental results are shown in Fig. 4f–h. The NMSE is 0.107 using only 45 output nodes. The current best experimental performance for this task in photonic RC with 50 nodes also reports an NMSE of 0.107[21] but with a slower processing speed of 0.9 MHz, and a larger physical footprint due to the use of a fiber-based system. Figure 4i highlights

the superior performance of our system in both operation speed and NMSE for the NARMA10 task, compared to existing physical RC systems through experiments and simulations[17,21,54–58].

## Extending to WDM operation

The implemented photonic NG-RC system can also be used for image classification. Here we demonstrate a two-class COVID-19 classification task[59]. The dataset comprises 2000 healthy X-ray samples and 2000 COVID-19 samples and each sample has 299 × 299 gray-scale pixels. We initially perform image preprocessing, including Fourier transform and low-pass filtering, to decrease input dimensions, as shown in Fig. 5a. The Fourier transform can be regarded as a form of unitary matrix operation[60]. Therefore, these preprocessing steps can be accomplished using photonic circuits[61] or simply an optical lens. The real part values of the eight lowest spatial frequencies from the transformed images are flattened and then serve as inputs to our photonic NG-RC chip for training and testing. We use 3,200 image samples as the training dataset, and 800 image samples as the testing dataset.

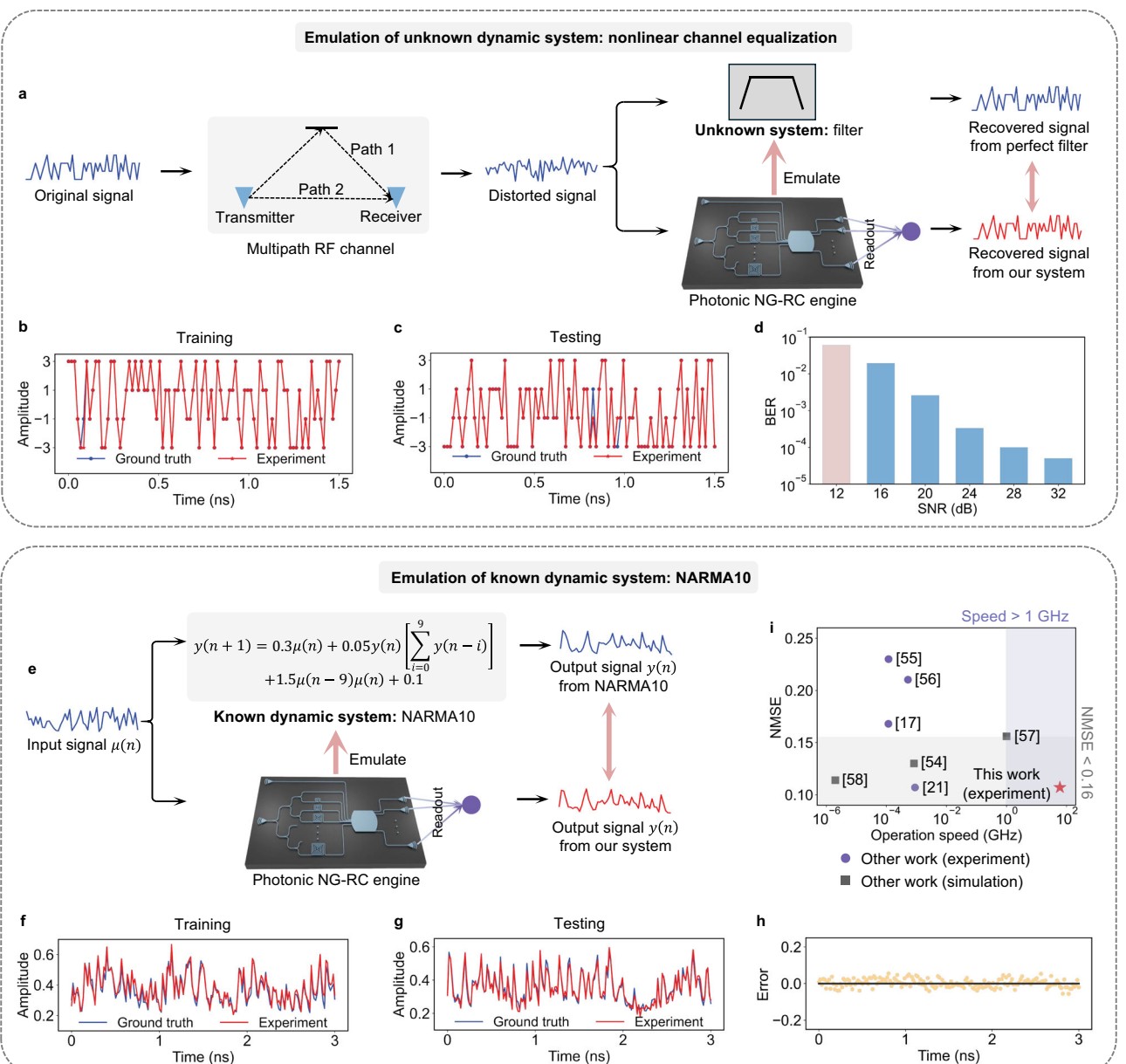

**Fig. 4 | Experimental demonstration of system emulation tasks. a** Concept of emulating an unknown filter using our photonic NG-RC system. **b**, **c** Experimental results during the training (**b**) and testing (**c**) phases in the NCE task. The reconstructed signal (red) and the original signal (blue) are plotted. **d** The experimental result shows a BER of 0.06 at an SNR of 12 dB, along with the simulated BER for different SNR levels. **e** Concept of emulation of a known dynamic system

(NARMA10) with our photonic NG-RC system. **f**, **g** Experimental results during the training (**f**) and testing (**g**) phases in the NARMA10 task. The generated sequence (red) and the target sequence (blue) in the NARMA10 task are plotted. **h** The error between the experimental results and ground truth during the testing phase. **i** Comparison of the performance on the NARMA10 task with other reported work, focusing on NMSE versus the operation speed.

We first use a single wavelength as the carrier for demonstration. The input signal is modulated onto a laser operating at a wavelength of 1550 nm and then transmitted to the photonic NG-RC system for processing. In the single wavelength case, similar to previous experiments, we collect output from 45 spatial output ports for data processing, as shown in Fig. 5b, and obtain a classification accuracy of 92.1%, as shown in Fig. 5d, which is higher than those reported in refs. 59,62.

We then use multiple wavelengths to demonstrate the scalability of our photonic NG-RC system enabled by WDM. In this experiment, five wavelengths (1548 nm, 1549 nm, 1550 nm, 1551 nm, and 1552 nm) are used to encode the same input signal. As the signals pass through the star coupler, each wavelength experiences a different transfer

function matrix, resulting in five distinct output signals as shown in Fig. 5e. The number of outputs determines the computational capacity. When the wavelength number is $P = 5$ and the number of the input vector is $N = 8$, according to the operation principle, the required spatial output port is $\frac{(N+1)(N+2)}{2P} = 9$.

To illustrate the capability of using WDM to extend computational capacity, we collect data from only nine spatial output ports instead of 45. If only a single wavelength is used, the classification accuracy is reduced to 73.1% with the confusion matrix shown in Fig. 5c. However, using five wavelengths improves the accuracy to 92.3% with an area under the curve (AUC) of 0.94, comparable to the case with a single wavelength but using 45 output ports. Therefore, employing WDM enhances the computing capacity of our system and reduces the

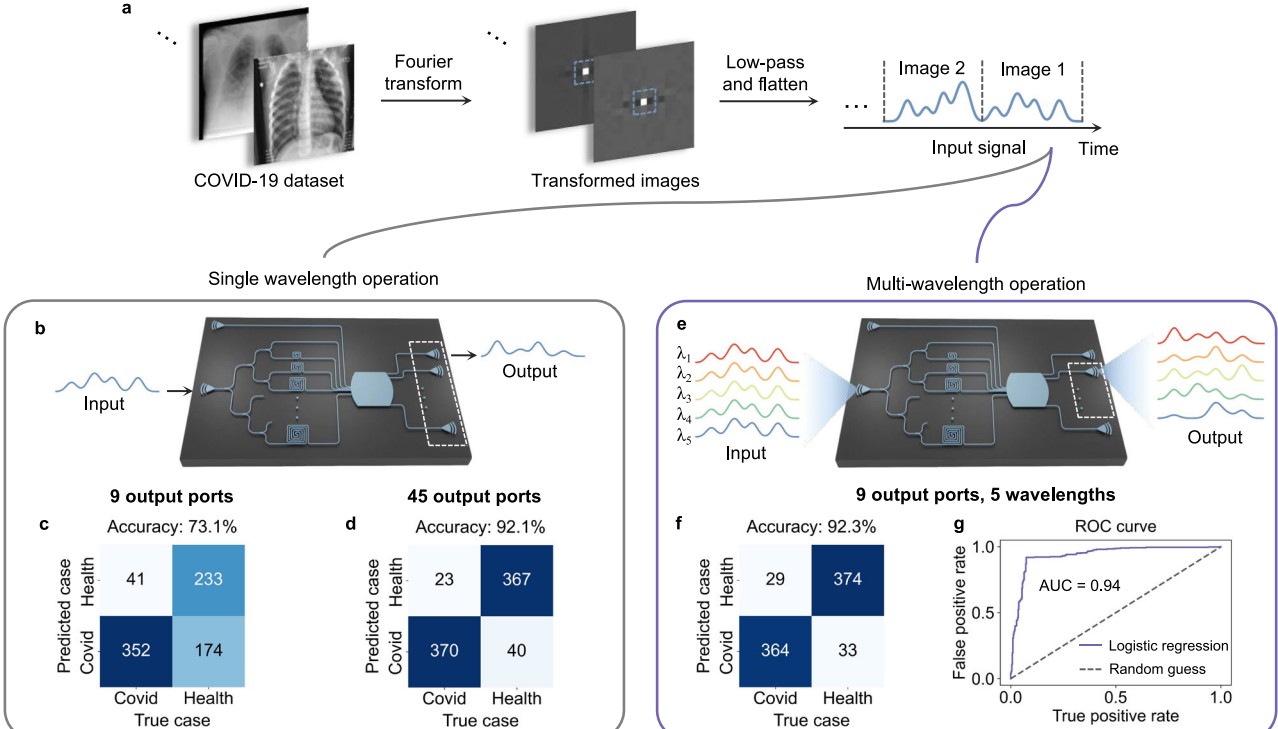

**Fig. 5 | Experimental demonstration of image classification task and WDM.**
**a** Image preprocessing for COVID-19 task. **b** Experimental scheme using a single wavelength. **c** Confusion matrix when using a single wavelength and 9 output ports. **d** Confusion matrix when using a single wavelength and 45 output ports. **e** Experimental scheme using multiple wavelengths. **f** Confusion matrix when employing WDM. **g** ROC curve when employing WDM.

number of required spatial output ports, thereby minimizing the system's footprint and improving scalability, especially when the output port number is limited by optical SNR.

## Discussion

**Feasibility of a fully integrated photonic NG-RC chip:** Silicon photonic platforms present a highly promising approach for the full integration of photonic NG-RC on a single chip, as depicted in Fig. 1c. Silicon photonic circuits with thousands of elements have already been successfully demonstrated[63,64]. Commercial foundries such as Advanced Micro Foundry (AMF) provide essential components for the full integration, including 56 GHz MZI modulators, 70 GHz photodetectors, and 220 nm-thick silicon waveguides for delay lines, star couplers, and microring weight banks. We have also previously demonstrated on-chip microring weight banks for matrix multiplication[8]. Here, we use microring weight banks to perform complex operations (as discussed in Supplementary Note 3).

Furthermore, WDM light sources can be realized by silicon nitride soliton frequency combs, which are compatible with CMOS technology and can generate hundreds of evenly spaced comb lines[9]. The soliton combs can be integrated with other photonic components by fiber and wire bonding, further enabling the full integration of our photonic NG-RC engine.

**Energy consumption:** Our all-optical NG-RC architecture offers high-speed data processing with low power consumption. In single-wavelength operation, the system can be viewed as performing three key operations: an $(N + 1) \times M$ complex matrix operation (via the star coupler), two $\frac{M}{2} \times 1$ complex matrix operations (through the microring weight banks), and two complex square operations (in the balanced photodetector), where $N$ is the number of delayed input copies and $M$ is the number of outputs. The system achieves ~211 Tera operations per second with $N = 8$, $M = 45(9 \times 10/2)$ (see Supplementary Note 5), and a line rate of 60 GBaud. The total power consumption is about 5.1

watts (see Supplementary Note 5), yielding an energy efficiency of ~41 TOPS/W—two orders of magnitude higher than the NVIDIA H100, which typically achieves 0.15 TOPS/W[12].

**Scalability:** Conventional RC uses randomly sampled matrices to define the underlying recurrent neural network. These matrices are typically not optimized. When translating such an RC algorithm into a physical system, this non-optimized configuration results in a clumsy system with unnecessary devices, leading to low computing density and difficulty in scaling up. In contrast, next-generation RC eliminates random recurrent connections, reducing the number of required devices and output ports. This allows us to implement its core function with a simple 0.04 mm² star coupler, significantly improving scalability in the spatial domain.

We compare the chip area per node with other photonic RC systems, where node count determines computational capacity. Our calculations include input ports and the reservoir layer, excluding the output layer, as it is a standard linear regression layer and most of the work is done offline on a computer. As shown in Supplementary Note 10, our system has the smallest chip area per node and uses the minimum number of outputs. Scaling to more nodes only slightly increases the star coupler size without changing its design. A 6.25 mm² star coupler can support over 5000 nodes, limited only by optical power for maintaining a good signal-to-noise ratio (discussed in Supplementary Note 7).

The number of computing nodes can be greatly increased using wavelength-division multiplexing, without expanding the star coupler or adding spatial output ports. Figure 5 shows how encoding inputs at parallel wavelengths scale up computing power. By leveraging wavelength-division multiplexing and an optical comb source, the number of computing nodes can be amplified hundreds of times.

Additionally, our system's ultra-high speed allows easy scaling through time-division multiplexing. We evaluate the number of effective nodes in 1 cm² and 1 μs across various schemes, as illustrated

in Supplementary Note 10. The results show that our system's node count, which determines the computational capacity, is up to four orders of magnitude higher than other approaches.

Implementation of higher-order polynomials: While a simple quadratic polynomial offers good computational power, certain problems may require higher-order polynomials. Our system can implement these higher-order polynomials by cascading modulators at the input stage, as illustrated in Supplementary Fig. 8. The input consists of a constant term $c$, a linear term $X$, and a quadratic term $X^2$, which are combined using a star coupler, weighted by a microring weight bank, and detected by a balanced photodetector. This configuration allows for the realization of third- and fourth-order terms, with the potential for implementing even higher-order terms by cascading additional modulators (see more details in Supplementary Note 9).

## Methods

### Chip design

The group index of these delay lines is ~4.24 at $\lambda = 1550$ nm by FDTD simulation. To achieve a delay of 16.7 ps in our design, the length difference of the neighboring delay lines is set as 1.18 mm (testing details are illustrated in Supplementary Note 2). The star coupler, with an area of $200\,\mu m \times 200\,\mu m$, has nine input ports and forty-five output ports. We designed the star coupler output port widths to ensure a nearly uniform power distribution across all ports. The widths decrease from $9.2\,\mu m$ at the edges to $2.6\,\mu m$ at the center. Each output port is followed by a $100\,\mu m$-long taper, transitioning into a single-mode waveguide with a width of 500 nm, as shown in Supplementary Fig. 1.

### Chip fabrication

The photonic NG-RC chip is fabricated on a silicon-on-insulator (SOI) wafer, as depicted in Supplementary Fig. 1. This SOI wafer has a $2.2\,\mu m$-thick oxide ($SiO_2$) cladding layer, a 220 nm-thick silicon (Si) layer, and a $2\,\mu m$-thick buried thermal oxide ($SiO_2$) layer. The width of the silicon waveguide of delay lines is 500 nm in our design. Our photonic NG-RC structure is composed of several integrated delay lines and a star coupler, as depicted in Supplementary Fig. 1.

### Data processing

Here, we offer some details on the data encoding and processing. For all tasks, the data is normalized to the range [-1, 1]. For the Lorenz63 task, we adopt the convention used in digital NG-RC[35], where the input data sent to our photonic NG-RC engine is uniformly sampled at intervals of five-time steps. Consequently, the following data points: $x_t$, $y_t$, $x_{t-5}$, $y_{t-5}$, $x_{t-10}$, $y_{t-10}$, $x_{t-15}$, and $y_{t-15}$ are combined as the input to our photonic NG-RC to predict $z_{t+1}$. In the NARMA10 task, we use the variable $\mu_t$ to predict $y_{t+1}$. We concatenate $\mu$ at time points $t$, $t-1$, $t-2$, $t-3$, $t-9$, $t-10$, $t-11$, and $t-12$, and use these data points as inputs to our system.

After collecting the reservoir states, we train a digital readout layer to map the outputs of our photonic NG-RC to targets. For the prediction and emulation tasks, we train a digital linear readout layer using Tikhonov Regularization, which helps prevent overfitting to the training data. The weight matrix $\mathbf{W}_{out}$ can be expressed as

$$\mathbf{W}_{out} = \mathbf{Y}\mathbf{X}^T(\mathbf{X}\mathbf{X}^T + \lambda\mathbf{I})^{-1} \qquad (10)$$

where $\mathbf{X}$ is the output of our photonic reservoir computing, $\mathbf{Y}$ is the target vector, $\lambda$ is a regularization parameter and $\mathbf{I}$ is the identity matrix. In our experiment, we use $\lambda = 1.4 \times 10^{-5}$.

The readout layer of the image classification task (COVID-19 task) is trained using logistic regression. Unlike linear regression, which predicts continuous values, logistic regression estimates the probability that the outputs of the RC belong to a specific class. This method applies the logistic function to the linear combination of the photonic NG-RC's outputs and uses a threshold to classify them into different classes.

## Data availability

The data generated in this study are provided in the Supplementary Information/ Source Data file. Source data are provided with this paper.

## Code availability

The codes that support the findings of this study are available from the corresponding author upon request.

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

## Acknowledgements

This work was supported by Innovation and Technology Fund (ITS/237/22, ITS/226/21FP); Research Grants Council, University Grants Committee (RGC ECS 24203724, RGC YCRF C1002-22Y, NSFC/RGC Joint Research Scheme N CUHK444/22); National Natural Science Foundation of China (62405258); Shun Hing Institute of Advanced Engineering (RNE-p4-22); Chinese University of Hong Kong (CUHK Direct Grant 170257018, 4055143).

## Author contributions

D.W. and C.H. conceived the ideas. D.W. and G.H. designed and simulated the system structure. D.W. designed the experiment. D.W. and Y.N. completed the experimental data collection and analyzed the results. D.W., Y.N. and C.H. wrote the manuscript. H.K.T. provided experimental advice and revised the manuscript. C.H. supervised the research and contributed to the general concept and interpretation of the results. All the authors contributed to the manuscript.

## Competing interests

The authors declare no competing interests.
