## [Transparent Peer Review file · Nature Communications]

Ultrafast Silicon Photonic Reservoir Computing Engine Delivering Over 200 TOPS

Corresponding Author: Professor Chaoran Huang

Version 0:

Reviewer comments:

Reviewer #1

(Remarks to the Author)

This article presents the first photonic NG-RC to be implemented on a photonic chip. The device uses a series of on-chip delay lines to introduce delayed copies of the input signal into a star coupler which mixes these inputs. The detected outputs are treated as the NG-RC output feature vector and used for inference. I think this is potentially a very attractive platform, but the work needs significant clarification in several areas.

At a high-level, this work makes claims about speed and power consumption that would benefit from clarification:

(1) Speed: This work emphasizes the computing speed of 60 GBaud and 100 TOPS/mm². These are impressive numbers and I agree that this approach has the potential to operate at very high data rates. However, the device presented in this work does not operate at 60 GHz. Instead, it relies on measuring and digitizing the output from 45 grating couplers and then applying output weights digitally. The gap between the passive device tested here and the projected performance of a future device should be made clear.

(2) Energy consumption: The article emphasizes that the chip itself is passive and does not consume energy—but using the chip requires a seed laser, modulator, 2 EDFAs, and a photodetector, as well the digital-to-analog converter and analog-to-digital converter. As a result, I did not find the energy comparison of “with” or “without” in Table 1 to be particularly helpful. I would find it more helpful to consider the total energy required to use this system (or projections for a future version of it), in comparison to a software, GPU-based implementation. From an energy-perspective, photonic computing systems tend to outperform digital electronics as the size of the input data increases... for the 8-element input vectors processed by this system, it's not clear how much of an advantage the photonic system would have after accounting for the overhead cost of converting to and from the optical domain.

Beyond these high-level comments about putting the claims in context, I had several specific comments and questions:

(3) Fig. 1 claims that this photonic NG-RC has an “exact equivalence” to a digital NG-RC. But in this case, each element in the output feature vector includes combinations of the constant, the linear feature vector, and various non-linear, quadratic mixing terms. This is very different from the digital NG-RC in which the linear and non-linear elements are independent. As a result, I'm not convinced the claim of “exact equivalence” is justified. Second, have you conducted any analysis or simulations to evaluate the relative computing power of these feature vectors? It may be that the feature vector produced by the photonic system here enables similar computing accuracy for some tasks, but I think the distinction should be made clear.

(4) Experimentally, Fig. 2 seems to imply that light from a single output waveguide was measured at a time. Was the output fiber moved sequentially to probe all 45 output grating couplers? The measurement procedure should be clarified at least in the supplement. The number of output waveguides should also be listed in the experimental setup instead of requiring the reader to refer to the Supplement, since this is such an important parameter.

(5) Environmental stability: while the chip itself is likely fairly stable, it seems that you would be sensitive to phase variations along the 2 optical fiber paths coupling the encoded information and the bias to the chip. How was this sensitivity handled and how long did it take to conduct a measurement?

(6) Can you describe the W matrix describing the transmission matrix of the star coupler? Is it effectively a fully random matrix?

(7) I found it interesting that the constant was introduced to the star coupler, as opposed to simply adding a constant in software, or as a pass-through waveguide skipping the star coupler in hardware. Can you comment on this decision and the

importance of adding this constant term to the star coupler?

(8) Lorenz task: Can you add some context for the NMSE achieved in the Lorenz task? Right now, the article merely says that this implies “good prediction ability”. How does this compare to a digital NG-RC or other photonic RCs?

(9) NARMA-10: The comparison in Fig. 4 is restricted to RCs with 50 nodes, but much lower NMSE has been achieved with more nodes. For example, Ref. 37 reported NMSE of 0.0225. Can this approach scale up to larger numbers of nodes to achieve state-of-the-art inference rates? For readers less familiar with this task, it would also help to mention the accuracy of some competing RCs with higher node-counts for context.

There were also a series of statements that require clarification or context:

(10) Meta-parameters: It seems odd to say that this approach does not involve meta-parameters in Table 1... the number of input delay lines is a meta-parameter, as is the type of mixing (i.e. the elements in W) and the number of output waveguides.

(11) Line 147: “the response of our RC is the same as the output of the NG-RC O_{total} ” is not accurate. The response here is impacted by the same terms, but you don’t have separate access to the linear and non-linear terms, which could be an important distinction for some tasks.

(12) Line 149: “values of $c, a,$ and b do not affect the final computing output...” I agree that this approach is relatively fabrication tolerant, but these coefficients still impact the computing output because they determine the type of mixing used to produce the O_{total} feature vector.

(13) Line 169: If you bias the EOM at the null point, you won’t be encoding linearly up to $|V| < V_{pi}/4$. This would introduce a non-linear encoding. Also—how were the negative values in the Lorenz task treated? Were they encoded as positive and negative voltages such that the same output power was encoded for positive and negative values?

(14) Line 239: Typo: “slowest” spatial frequencies

Reviewer #2

(Remarks to the Author)

In this paper, the main claim of the authors consists in the optical implementation of a high-speed Next Generation Reservoir Computing (RC) system on an integrated photonic chip. On the base of a passive system, they assert the superior potential of the proposed integrated next generation RC architecture in terms of classification and prediction performance based on standard tasks operating at an estimated speed of 60 Gbaud and a computational density of 102 TOPS/mm². The main architecture is based on a star coupler with on-chip coupler, limited to 8 delays in the proposed study. Whereas implementing Next Generation RC on an integrated photonic chip would have several advantages even over conventional optical RC, namely a significantly shorter learning length, and fewer hyperparameters, it is not clearly demonstrated in the article how the proposed architecture offers changeability and reliability and prospects of scalability over other optical RC implementation or even digital NGRC with respect to large-scale chaotic systems. This paper might be highly interesting and even it presents experimental results with high-speed, it does not demonstrate the advantages of the proposed passive architecture in comparison with other optical NGRC and optical conventional RC regarding the task treated (see ref [20-28] and also arXiv [35,36]). In fact, because of the passive nature of the star coupler, it is not possible to adjust critical parameters such as the weight and the non-linear activation function, which constitute a clear limitation to the proposed method to solve more complex tasks. This is the major criticism here, of course, there is an integrated device that can process with high-speed, but only part of it is on chip and is passive (what about the EO modulator, EDFA, laser and Photodiode...). The manuscript is well written, well supported by a comprehensive review of the literature in the field, but it does not offer enough information to clearly specify the important features related to the experimental set-up as well as the critical aspects related to it. The paper does not propose a convincing advancement in the field of optical computing and it is my recommendation that the paper cannot be accepted for publication.

Reviewer #3

(Remarks to the Author)

Report on A 103-TOPS/mm² Integrated Photonic by D. Liang et al.

In this paper the authors report an integrated optics implementation of a “Next Generation Reservoir Computer”. The paper is innovative, and will interest the community working on photonic neuromorphic computing. The integrated optics system reported seems rather standard. The tasks addressed are not amongst the most challenging. On the other hand the speed achieved is impressive. I have a number of substantial remarks that make it difficult to give a definitive judgement on the paper. I would need to see a revised version to give a definitive opinion, although I tend to think it will be a bit below the level sought for in Nature Communications.

I have some serious reservations about the title and some claims in the paper. Indeed the authors implement off chip significant parts of their information processing system. The most challenging part, from a photonics point of view, would be to implement the photodetection on chip and to put weights optically or electronically on chip. Doing so could considerably complicate the system. In addition the authors outsource much of the computation using a high end oscilloscope to measure the signals and then digital postprocessing. Overall the authors need to do a fairer evaluation of the performance of their system. What would it look like if the detection part was done on chip? Given the overhead of converting electrical signals to optical, and then optical to electrical followed by digitization, how would the system compare to a fully electronic system? For these reasons the title, the “computing density” discussed in the paper (see e.g. the abstract), and Table 1, all seem to me much too optimistic. Overall I feel that Table 1 is quite misleading, omitting some important papers, and counting energy consumption for instance in a very weird way.

Concerning terminology, the name “next generation reservoir computing” is a bit pompous. In fact, an alternative way of looking at the experiment is as an “Extreme Learning Machine” (another pompous name) that is adapted to process a time

series by feeding it with several delayed inputs. I recommend the authors connect with this alternative interpretation.

The "next generation reservoir computing" is also closely related to

L. Jaurigue, E. Robertson, J. Wolters, and K. Lüdge, "Reservoir computing with delayed input for fast and easy optimisation," *Entropy* 23, 1560 (2021).

L. Jaurigue and K. Lüdge, "Reducing reservoir computer hyperparameter dependence by external timescale tailoring," *Neuromorphic Comput. Eng.* (2024).

in which several delayed inputs are fed into a standard reservoir computer. I recommend the authors connect with these works.

I find the approach of the authors very close, in its principle, to some other optical extreme learning machines, as well as to Ref 6 "11 tops photonic convolutional accelerator." A discussion of this connection, as well as to other works that implement photonic information processing using linear optics followed by photodetection, is called for. Given the way the authors process the time series, this comparison seems very relevant (even if Ref 6 and some other papers do not deal with time series). In fact the third task the authors consider is not based on a time series, and thus clearly falls into the "extreme learning machine" category mentioned above.

Supplementary material: "Our experiment has nine input data (including constant) and thirty-six corresponding quadratic polynomials. Thus, the star coupler has forty-five output ports in total."

Please explain this important consideration in more detail. Why 45? (I think I know, but this needs explaining). I would put this important consideration in the main text in section 2.2.

In fact it would be very interesting to understand how many output ports are really needed for good performance. Could the authors give a plot of performance as a function of the number of output ports that are used? What happens if less than 45 output ports are used? Any idea whether using more than 45 output ports could be beneficial?

What if the task requires more than quadratic polynomials in the input? Could the system be adapted to this case?

Fig. 1 is a bit cramped vertically. I don't understand the text "exact equivalence" in the figure.

End of section 1. "Metaparamters" : missing E

"and providing interpretable results » not clear what this means.

Section 2, after Eq. 1 "the quadratic polynomial feature vector has been demonstrated good prediction » garbled.

After Eq. 2 "The neighboring delay line introduces a time delay of Δt ," Probably garbled (check meaning and plurals). There are 8 delay lines.

End of section 2.1 "Importantly, the actual values of the c_i , a_i , k , and $b_{i,p,q}$ do not affect the final computing output because the final readout layer can adjust them to the optimal values through training." This is wrong, except if n is large enough (and even when n is large, there are values of c , a , b for which there will be a problem). Please check what you mean.

Section 2.3.1 Lorenz system.

Eq. 6. Please specify the time step you use, as this defines the complexity of the task.

Fig 3b: I don't understand why this picture is informative. Indeed, for one step ahead prediction, except if the prediction is very bad, one will reproduce approximately the Lorenz attractor. Please clarify or remove.

Fig 3c and d: why is the horizontal axis in ns? There are no time units in Eq. 6 (see also remark about the time step used). The horizontal axis is probably the real time used to input the data, but this is not explained.

Below eq. 7, the NMSE for the Lorenz task s_i reported. Please compare with what has been achieved previously in the literature.

Finally the english is often a bit weak, with some sentences difficult to understand (I have flagged some examples above). Please check thoroughly throughout.

Version 1:

Reviewer comments:

Reviewer #1

(Remarks to the Author)

The authors did a nice job of addressing my concerns and clarifying this work. I appreciated the updated energy analysis and the conceptual figure 1 as well as the clarification of the role the constant power waveguide and the equivalence of the feature vector to a digital NG-RC feature vector. I recommend it for publication.

One typo I noticed: section 4.3 should be data processing, not "date" processing.

Reviewer #2

(Remarks to the Author)

After a careful reading of the authors' responses and the numerous supplements provided, including experimental results, the manuscript proposes content that could be of interest to readers of Nature Communications, also considering the major changes proposed (new title, new figures to present the WDM experiments, etc.). I recommend accepting the manuscript for publication in the journal Nature Communications.

Reviewer #3

(Remarks to the Author)

Report on Ultrafast Silicon Photonic Reservoir Computing
2 Engine Delivering Over 200 TOPS

The authors have significantly improved the manuscript and addressed the referee comments. I believe the manuscript can be published in Nature Communications with a additional changes.

1)

Figure 1 (b) connection between “digital input x” and “Input: N+1 terms”. I would suggest trying to indicate that these connections are all of different lengths, inducing different delays. This is very clear in panel c, but missing in panel b.

2)

Figure 2 and Section 2.5. In Fig 2, is the Band Pass Filter used in normal operation when a single laser is used? If so why? On the other hand when multiple lasers are used (section 2.5) the BPF may be essential.

I would appreciate having (possibly in the supplementary material) a figure illustrating the experimental setup when multiple wavelengths are used (as it probably differs slightly from Fig 2), and additional explanations on how the system is operated in this case. Are the multiple lasers used at the same time, or sequentially? What BPF is used?

3) Energy Consumption and Supplementary Note 4. I do not understand how the authors take into account the energy consumption in the system they use. For instance the Fast Oscilloscope consumes approximately 1kW (not 5.1 W). I believe the authors should include a comparison of: 1) energy consumption in the system they implemented (including consumption of AWG, oscilloscope, current sources, laser driver, etc.); 2) energy consumption of a realistic system with accessible components.

Similarly, does Fig S3 refer to the real experimental system, or to a future system? Please clarify.

The supplementary material contains a new table that is illustrated by figures taken from a dozen publications. The authors need to check that they have authorization to publish these figures.

Dear Reviewers,

Thank you very much for taking your time to review our manuscript titled "A 103-TOPS/mm² Integrated Photonic Computing Engine Enabling Next-Generation Reservoir Computing" (Manuscript ID: NCOMMS-24-30861). We greatly appreciate the valuable comments and suggestions provided by you and the reviewers. We have carefully revised the manuscript and the supplementary document accordingly. Detailed point-by-point responses are provided below.

Response To Reviewer 1 (R1)

Comment 1

This article presents the first photonic NG-RC to be implemented on a photonic chip. The device uses a series of on-chip delay lines to introduce delayed copies of the input signal into a star coupler which mixes these inputs. The detected outputs are treated as the NG-RC output feature vector and used for inference. I think this is potentially a very attractive platform, but the work needs significant clarification in several areas.

Response 1

We are grateful for your thorough review of our manuscript. Your positive comments and constructive feedback are very valuable to us. Thank you for your support and encouragement.

Comment 2

This article presents the first photonic NG-RC to be implemented on a photonic chip. The device uses a series of on-chip delay lines to introduce delayed copies of the input signal into a star coupler which mixes these inputs. The detected outputs are treated as the NG-RC output feature vector and used for inference. I think this is potentially a very attractive platform, but the work needs significant clarification in several areas.

At a high-level, this work makes claims about speed and power consumption that would benefit from clarification:

(1) Speed: This work emphasizes the computing speed of 60 GBaud and 100 TOPS/mm². These are impressive numbers and I agree that this approach has the potential to operate at very high data rates. However, the device presented in this work does not operate at 60 GHz. Instead, it relies on measuring and digitizing the output from 45 grating couplers and then applying output weights digitally. The gap between the passive device tested here and the projected performance of a future device should be made clear.

Response 2

Thank you for your question regarding computing speed.

In our work, we demonstrate an information throughput of 60 Gbaud. The reviewer is correct that the output layer is currently implemented digitally and processed offline. However, it is feasible to implement the output layer entirely optically using programmable MZI arrays, PIN attenuators, or MRR weight banks [1, 2, 3]. This would allow real-time weighting at the output layer, eliminating speed constraints.

Following the reviewer's suggestion, we have revised the manuscript to include an outlook for a system with a fully integrated readout layer. Adding the optical readout layer would require only a single optical output digitizer, a significant improvement over the 45 outputs needed in the current setup. Meanwhile, we also project the hardware performance in terms of energy

efficiency with the all-optical engine. The details will be discussed in the following response.

Revision 2

1. **Figure 1** in our manuscript is revised to show the design chip including the optical readout layer using MRR arrays.
 2. We add the working principle of the optical readout layer between lines 147 and 167.
 3. We estimate the energy consumption of the system, including the optical readout layer and peripheral components such as the laser, DAC, modulator, EDFA, PD, and ADC. Details are provided in the Discussion section (lines 373-383) and Supplementary Note 4. A comparison between our optical engine and the GPU H100 is also included in Supplementary Note 4.
-

Comment 3

(2) Energy consumption: The article emphasizes that the chip itself is passive and does not consume energy—but using the chip requires a seed laser, modulator, 2 EDFAs, and a photodetector, as well the digital-to-analog converter and analog-to-digital converter. As a result, I did not find the energy comparison of “with” or “without” in Table 1 to be particularly helpful. I would find it more helpful to consider the total energy required to use this system (or projections for a future version of it), in comparison to a software, GPU-based implementation. From an energy-perspective, photonic computing systems tend to outperform digital electronics as the size of the input data increases... for the 8-element input vectors processed by this system, it’s not clear how much of an advantage the photonic system would have after accounting for the overhead cost of converting to and from the optical domain.

Response 3

Thank you for your valuable suggestions on improving the concise presentation of our results.

In our original manuscript, we aim to emphasize that the reservoir layer in our work only requires a passive device (a simple star coupler), in contrast to most prior research that requires several active devices such as lasers and modulators, which constrain computing speed and increase energy consumption.

However, we agree with the reviewer that simply differentiating our work using the terms "with" and "without" energy consumption is not precise, as the peripheral circuits for signal generation and detection indeed require energy. We appreciate the reviewer’s good suggestion for more accurately presenting the energy consumption advantages of our work. In our revised manuscript, we calculate the energy consumption of our optical engine (the projected version with an optical readout layer) including the peripheral circuits for signaling. The energy has been compared with the state-of-the-art GPU H100.

Our optical engine is composed of the following key components: a PPCL500 laser (1 W), an EDFA (0.35W)[4], a 240 GSamples/s DAC and an on-chip modulator (about 0.75 W)[5,6], ninety metal heaters for the output layer (approximately 2.25 W), and two on-chip

photodetectors and a 256 GSamples/s ADC (about 0.75 W)[7]. Therefore, the optical engine, operating at a line rate of 60 Gbaud (the same as our experiments) and performing 211 Tera operations per second, has an energy efficiency of around 41TOPS/W (see Supplementary Note 4). In contrast, NVIDIA’s H100 achieves an energy efficiency of roughly 0.15 TOPS/W cited from [8]. Significantly, as shown in **Figure R1.3**, when the reservoir size further increases, our optical engine offers an even more substantial energy consumption advantage over GPUs when processing large-scale NG-RC.

Figure R1.3 The energy consumption per operation of NG-RC with different input vector sizes completed by our optical engine and GPU H100.

Revision 3

1. We delete **Table 1** from the original manuscript and replace it with a new table in Supplementary Note 9 that provides a performance comparison with previous photonic reservoir computing systems. The comparison covers metrics such as speed, computing density, nonlinear node count, footprint, and energy consumption for performing the same task.
2. We add the energy consumption of the system, including the optical readout layer and peripheral components such as the laser, DAC, modulator, EDFA, PD, and ADC. Details are provided in the Discussion section (lines 373–383) and Supplementary Note 4. A comparison between our optical engine and the GPU H100 is also included in Supplementary Note 4.

Comment 4

Beyond these high-level comments about putting the claims in context, I had several specific comments and questions:

(3) Fig. 1 claims that this photonic NG-RC has an “exact equivalence” to a digital NG-RC. But in this case, each element in the output feature vector includes combinations of the constant, the linear feature vector, and various non-linear, quadratic mixing terms. This is very different from the digital NG-RC in which the linear and non-linear elements are independent. As a result, I’m not convinced the claim of “exact equivalence” is justified.

Second, have you conducted any analysis or simulations to evaluate the relative

computing power of these feature vectors? It may be that the feature vector produced by the photonic system here enables similar computing accuracy for some tasks, but I think the distinction should be made clear.

Response 4

Thank you for your suggestion. We have removed the term "exact equivalence" to avoid confusion and replotted **Figure 1** in the revised manuscript.

We have revised the manuscript to explain why our system functions equivalently to the NG-RC:

After the star coupler, the output vector $\mathbf{y}_{\text{star},\lambda}$ is expressed as:

$$\mathbf{y}_{\text{star},\lambda} = \mathbf{W}_{\text{star},\lambda} \cdot (\mathcal{C} \oplus \mathbf{X})$$

Here, $\mathbf{W}_{\text{star},\lambda}$ denotes an $M \times (N + 1)$ complex matrix, representing the transfer function of the star coupler at wavelength λ , where N is the number of delayed input copies and M is the dimension of the output vector.

The outputs from the star coupler are followed by a readout layer, which can be implemented either digitally or optically. In a digital implementation, each output of the star coupler is detected by a photodetector, producing an output given by:

$$\begin{aligned} y_{PD,i} &= |\mathbf{W}_{\text{star},\lambda,i} \cdot (\mathcal{C} \oplus \mathbf{X})|^2 \\ &= \underbrace{c_i}_{\text{constant}} + \underbrace{\sum_{n=1}^N a_{n,i} x_{t+1-n}}_{\text{linear terms}} + \underbrace{\sum_{m=1}^N \sum_{n=m}^N b_{mn,i} x_{t+1-m} x_{t+1-n}}_{\text{quadratic terms}} \end{aligned}$$

where $y_{PD,i}$ is the output of the photodetector. c_i , $a_{n,i}$, and $b_{mn,i}$ are constants, which are determined by the transfer function of the star coupler. This equation shows that the output of each photodetector is a combination of a constant, linear inputs, and their quadratic polynomials. M outputs form a vector \mathbf{y}_{PD} , where each element is a similar mixture but with different coefficients determined by the transfer function matrix of the star coupler. When the outputs number $M \geq \frac{(N+1)(N+2)}{2}$, \mathbf{y}_{PD} can be transformed into the feature vector required by the NG-

RC, consisting of constant, linear, and nonlinear components, using a linear matrix. This means there exists a \mathbf{W} such that $\mathbf{W} \cdot \mathbf{y}_{PD} = \mathcal{C} \oplus \mathcal{O}_{\text{linear}} \oplus \mathcal{O}_{\text{nonlinear}}$, where $\mathcal{C} \oplus \mathcal{O}_{\text{linear}} \oplus \mathcal{O}_{\text{nonlinear}}$ is the feature vector in the digital NG-RC. In this case, the system functions equivalently to the NG-RC.

In our chip, the number of input ports is 9, 8 for delayed input copies, and 1 for the constant term. And the output number is $45=(9 \times 10)/2$. Therefore, the feature vectors from the star coupler have the same computational power as the feature vector in the digital NG-RC.

Revision 4

We have rewritten the principle to explain why our system has the same computational function as the digital NG-RC (lines 136-143).

Comment 5

(4) Experimentally, Fig. 2 seems to imply that light from a single output waveguide was measured at a time. Was the output fiber moved sequentially to probe all 45 output grating couplers? The measurement procedure should be clarified at least in the supplement. The number of output waveguides should also be listed in the experimental setup instead of requiring the reader to refer to the Supplement, since this is such an important parameter.

Response 5

The output was measured sequentially at the 45 output grating couplers. We have added the measurement procedure and added the number of outputs in the experimental setup section 2.2. Thank you for the suggestion.

Revision 5

1. Measurement procedure on lines 228-232

The optical outputs are coupled out and detected sequentially using an off-the-shelf photodetector (COHERENT, model XPDV3120R-VM-FA, with a 70 GHz bandwidth) and digitized by a real-time oscilloscope (KEYSIGHT, model UXR0592AP, with a 59 GHz bandwidth and 256 GSa/s sampling rate).

2. Add the number of output waveguides on lines 199-201

The star coupler occupies a footprint of 0.04 mm^2 , with 9 inputs (one for unmodulated light and eight for delayed signal copies) and 45 (i.e., $(9+1) \times 9 / 2$) outputs to generate the necessary feature vector.

Comment 6

(5) Environmental stability: while the chip itself is likely fairly stable, it seems that you would be sensitive to phase variations along the 2 optical fiber paths coupling the encoded information and the bias to the chip. How was this sensitivity handled and how long did it take to conduct a measurement?

Response 6

The reviewer raised a valid point regarding stability. While the two optical fiber paths introduce phase variance, this has a negligible effect on the computing accuracy because the phase variance occurs at a much slower timescale than the computing output.

We conduct an experiment to investigate whether the phase variance can affect the system performance, as shown in **Figure R1.6**. We observe output power varies over a timescale of KHz caused by the phase variance. However, when zooming in to the GHz timescale, the output power remains constant, confirming that the phase variance has a negligible effect on the computing output.

Figure R1.6 Experimental demonstration of environmental stability. The blue lines represent the experimental data during task execution, while the yellow lines correspond to the results from measuring environmental stability.

Revision 6

We add the discussion of system stability in Supplementary Note 5.

Comment 7

(6) Can you describe the W matrix describing the transmission matrix of the star coupler? Is it effectively a fully random matrix?

Response 7

Mathematically, any full-rank matrix would suffice. However, in designing the star coupler, we take into account two factors: 1. To ensure that the output power at the edge of the star coupler maintains a reasonable signal-to-noise ratio (SNR), we designed the width of the output ports to decrease quadratically from the edge to the center. 2. We carefully control the overall width of the star coupler to achieve both a high output power and a reasonable SNR for all outputs, as shown in **Figure R1.7**.

Revision 7

1. We revise **Figure 2** in our manuscript to incorporate the two figures below (**Figure R1.7**).
2. We clarify the requirements for the transmission matrix of the star coupler, along with the OSNR considerations that inform our design of the star coupler, on lines 201-208.

As discussed in the Principle section, mathematically, any full-rank transmission matrix from the star coupler would suffice. However, considering a lower power density at the edges compared to the center, we design the widths of the output ports to decrease quadratically from the edge to the center, as shown in Fig. 2c. This optimized design ensures that each output maintains a nearly uniform power distribution and SNR across all ports, as shown in Fig. 2d.

The light transmission within the star coupler is simulated using Finite-Difference Time-Domain (FDTD) and shown in Fig.2b.

Figure R1.7 a, Simulated normalized transmission for each output port under uniform and optimized distribution conditions. b, Simulated electric field distribution in the star coupler region by FDTD.

Comment 8

(7) I found it interesting that the constant was introduced to the star coupler, as opposed to simply adding a constant in software, or as a pass-through waveguide skipping the star coupler in hardware. Can you comment on this decision and the importance of adding this constant term to the star coupler?

Response 8

According to the NG-RC framework, it is important to establish the constant term, together with the first-order terms, and the second-order terms of the input vectors.

Equation (1) describes the situation when we introduce the constant term into the star coupler via a waveguide, where c represents the constant term, and x refers to any input vector element. This constant term interacts with the input vector element in the amplitude domain within the star coupler. After photodetection, the signal becomes a linear combination of the constant (c^2), the first-order term ($2cx$), and the second-order term (x^2)

$$(c + x)^2 \rightarrow c^2 + 2cx + x^2 \quad (1)$$

$$c + (x)^2 \rightarrow c + x^2 \quad (2)$$

Equation (2) presents the output if we only use a constant defined in software or a pass-through waveguide that skips the star coupler. In this case, the output will not contain the first-order term ($2cx$) which is important to NG-RC. Therefore, it is essential to introduce the constant term into the star coupler via a waveguide.

Comment 9

(8) Lorenz task: Can you add some context for the NMSE achieved in the Lorenz task? Right now, the article merely says that this implies “good prediction ability”. How does this compare to a digital NG-RC or other photonic RCs?

Response 9

The conclusion that our approach has strong predictive capabilities is supported by comparisons with prior works. The Lorenz task we do is not a simple one-step-ahead prediction like in most RC works [9,10]. Instead, we predict the z-axis information using the x and y-axis data from the Lorenz system. This cross-prediction is significantly more challenging than a one-step-ahead prediction because we lack prior information about the z-axis. This cross-prediction task is also done in [11], where an NMSE reported 0.009 but using 2000 output nodes. In contrast, our experiment achieves an NMSE of 0.0143 with only 45 nodes. This NMSE can be further reduced by employing a modulator with a higher bandwidth (currently, we use a 40 GHz modulator to generate a 60 Gbaud signal). When simulating the same operation digitally, without the constraints of bandwidth limitations and noise, the NMSE can be reduced to 3×10^{-4} with 45 nodes.

In addition, compared to the system in [11], our optical engine significantly outperforms in terms of speed and energy efficiency. Specifically, the system in [11] requires 0.1 seconds for each prediction and consumes about 1.45 J of energy per prediction. In comparison, our projected optical engine with optical weighting performs each prediction in just 16.7 ps and consumes only 85 pJ. This demonstrates that our optical engine offers superior computational speed and energy efficiency.

Revision 9

We add some context about the NMSE of the Lorenz task on lines 277-282.

Comment 10

(8) NARMA-10: The comparison in Fig. 4 is restricted to RCs with 50 nodes, but much lower NMSE has been achieved with more nodes. For example, Ref. 37 reported NMSE of 0.0225. Can this approach scale up to larger numbers of nodes to achieve state-of-the-art inference rates? For readers less familiar with this task, it would also help to mention the accuracy of some competing RCs with higher node-counts for context.

Response 10

Thank you for your suggestions. We have analyzed the relationship between NMSE and output number, as shown in **Figure R1.10**. The NMSE decreases almost exponentially with an increasing number of ports. When the port number reaches 210, the NMSE aligns with the results in Ref. 37 (the input port number needs to be increased to 20).

However, it is important to highlight an advantage of next-generation RC: it requires fewer feature vectors to achieve optimal performance. In our work, the required port number to achieve an NMSE of 0.0225 is 210, while in Ref. 37, it is 400. Additionally, we compare our results with other studies on NARMA-10, and the results consistently show that we can achieve a much lower NMSE with fewer number of outputs, as shown in **Figure R1.10**. This translates to reduced energy consumption and a smaller chip footprint.

Figure R1.10 NMSE of NARMA10 task with increasing number of outputs, when the input dimension changes accordingly with the number of outputs. The blue circles are the simulation results of our work, and the purple squares are the simulation results from related works.

Revision 10

1. We add **Figure R1. 10** in Supplementary Note 7 and explain the results.

Comment 11

There were also a series of statements that require clarification or context:
 (10) Meta-parameters: It seems odd to say that this approach does not involve meta-parameters in Table 1... the number of input delay lines is a meta-parameter, as is the type of mixing (i.e. the elements in W) and the number of output waveguides.

Response 11

Thank you for your questions. What we mean is that our approach does not require meta-parameters that demand careful optimization or extensive learning. For instance, it is straightforward to see that increasing the number of input delay lines N improves performance, while the number of outputs is determined by the relationship $(N+1)(N+2)/2$ based on the input. In contrast, many previous works lack clear guidelines for determining optimal meta-parameters, therefore requiring much more complex learning processes.

Revision 11

We **delete Table 1** from the original manuscript, as we realize several descriptions in the table are not precise. We replace **Table 1** with a new table in Supplementary Note 9 that provides a performance comparison with previous photonic reservoir computing systems. The comparison covers metrics such as speed, computing density, nonlinear node count, footprint, and energy consumption for performing the same task.

Comment 12

(11) Line 147: “the response of our RC is the same as the output of the NG-RC O_{total} ” is not accurate. The response here is impacted by the same terms, but you don’t have separate access to the linear and non-linear terms, which could be an important distinction for some tasks.

Response 12

We agree with your comments that the response of our RC is not exactly the same as the NG-RC. However, as we explained in **Response 4**, our system functions equivalently to the NG-RC.

We have rewritten the principle to explain why our system has the same computational function as the digital NG-RC (lines 136-143).

Comment 13

(12) Line 149: “values of c,a, and b do not affect the final computing output...” I agree that this approach is relatively fabrication tolerant, but these coefficients still impact the computing output because they determine the type of mixing used to produce the O_{total} feature vector.

Response 13

Using our system, the i -th output of the star coupler after photodetection is

$$\begin{aligned} y_{PD,i} &= |\mathbf{W}_{star,\lambda,i} \cdot (C \oplus \mathbf{x})|^2 \\ &= \underbrace{c_i}_{\text{constant}} + \underbrace{\sum_{n=1}^N a_{n,i} x_{t+1-n}}_{\text{linear terms}} + \underbrace{\sum_{m=1}^N \sum_{n=m}^N b_{mn,i} x_{t+1-m} x_{t+1-n}}_{\text{quadratic terms}} \end{aligned}$$

The equation shows the output is a mixture of constant, linear inputs and quadratic polynomials of inputs.

M outputs form a vector \mathbf{y}_{PD} , where each element is a similar mixture but with different coefficients. These coefficients are determined by the transfer function of the star coupler. When the output port number $M \geq \frac{(N+1)(N+2)}{2}$, \mathbf{y}_{PD} can be transformed into the feature vector required by the NG-RC, consisting of constant, linear, and nonlinear components, using a linear matrix. This means there exists a \mathbf{W} such that $\mathbf{W} \cdot \mathbf{y}_{PD} = C \oplus O_{linear} \oplus O_{nonlinear}$, where $C \oplus O_{linear} \oplus O_{nonlinear}$ is the feature vector in the digital NG-RC. In this case, the system functions equivalently to the NG-RC. Although variations in the transfer function of the star coupler result in different matrices \mathbf{W} , we can always determine the optimal weights for the readout layer through training, ensuring performance that is exact to that of the digital NG-RC. Therefore, our system is highly tolerant to fabrication variances.

Revision 13

1. We revise the principle of our system to clarify why our system has the same computational function as the digital NG-RC (details in **Revision 4** and **12**).
 2. We add a paragraph to explain the reasons behind the high fabrication tolerance of our system on lines 143-146.
-

Comment 14

(13) Line 169: If you bias the EOM at the null point, you won't be encoding linearly up to $|V| < V_{\pi}/4$. This would introduce a non-linear encoding. Also—how were the negative values in the Lorenz task treated? Were they encoded as positive and negative voltages such that the same output power was encoded for positive and negative values?

Response 14

The modulator we use is configured in a push-pull arrangement. When we bias the modulator at null points, the optical field after modulation can be written as:

$$E_{out} = E_{in} \sin\left(\frac{\pi V}{2 V_{\pi}}\right)$$

where E_{in} and E_{out} are the optical fields before and after the modulator, respectively. V is the voltage applied in the modulator. For $|V| < V_{\pi}/4$, the corresponding E_{out} can be approximated by a linear function written as:

$$E_{out} = \frac{\pi E_{in}}{2 V_{\pi}} V$$

where E_{out} is linearly proportional to V . Therefore, we use $|V| < V_{\pi}/4$ to linearly encode the input data. When we apply voltages of V and $-V$, we obtain the same light intensity. However, the laser amplitude E_{out} values are opposite.

Revision 14

We add the principle of amplitude modulation in Supplementary Note 1.

Comment 15

(14) Line 239: Typo: "slowest" spatial frequencies

Response 15

Thank you for spotting the typo. We have corrected it to "lowest spatial frequencies."

Reference

1. Shen, Yichen, et al. "Deep learning with coherent nanophotonic circuits." *Nature photonics* 11.7 (2017): 441-446.
2. Ashtiani, Farshid, Alexander J. Geers, and Firooz Aflatouni. "An on-chip photonic deep neural network for image classification." *Nature* 606.7914 (2022): 501-506.

3. Huang, Chaoran, et al. "A silicon photonic–electronic neural network for fibre nonlinearity compensation." *Nature Electronics* 4.11 (2021): 837-844.
4. Feng, Hanke, et al. "Integrated lithium niobate microwave photonic processing engine." *Nature* 627.8002 (2024): 80-87.
5. Kossel, Marcel A., et al. "8.3 An 8b DAC-based SST TX using metal gate resistors with 1.4 pJ/b efficiency at 112Gb/s PAM-4 and 8-Tap FFE in 7nm CMOS." 2021 IEEE International Solid-State Circuits Conference (ISSCC). Vol. 64. IEEE, 2021
6. Wang, Cheng, et al. "Integrated lithium niobate electro-optic modulators operating at CMOS-compatible voltages." *Nature* 562.7725 (2018): 101-104.
7. Khairi, Ahmad, et al. "A 1.41-pJ/b 224-Gb/s PAM4 6-bit ADC-based SerDes receiver with hybrid AFE capable of supporting long reach channels." *IEEE Journal of Solid-State Circuits* 58.1 (2022): 8-18.
8. Xu, Zhihao, et al. "Large-scale photonic chiplet Taichi empowers 160-TOPS/W artificial general intelligence." *Science* 384.6692 (2024): 202-209.
9. Köster, Felix, et al. "Data-informed reservoir computing for efficient time-series prediction." *Chaos: An Interdisciplinary Journal of Nonlinear Science* 33.7 (2023).
10. Cai, Xinyi, et al. "Scalable photonic reservoir computing based on pulse propagation in parallel passive dispersive links." *Applied Optics* 63.22 (2024): 5785-5791.
11. Wang, Hao, et al. "Optical next generation reservoir computing." *arXiv preprint arXiv:2404.07857* (2024).
12. Vinckier, Quentin, et al. "High-performance photonic reservoir computer based on a coherently driven passive cavity." *Optica* 2.5 (2015): 438-446.
13. Liang, Xiangpeng, et al. "Rotating neurons for all-analog implementation of cyclic reservoir computing." *Nature communications* 13.1 (2022): 1549.
14. Appeltant, Lennert, et al. "Information processing using a single dynamical node as complex system." *Nature communications* 2.1 (2011): 468.
15. Appeltant, Lennert, et al. "Constructing optimized binary masks for reservoir computing with delay systems." *Scientific reports* 4.1 (2014): 3629.

Response To Reviewer 2 (R2)

Comment 1

In this paper, the main claim of the authors consists in the optical implementation of a high-speed Next Generation Reservoir Computing (RC) system on an integrated photonic chip. On the base of a passive system, they assert the superior potential of the proposed integrated next generation RC architecture in terms of classification and prediction performance based on standard tasks operating at an estimated speed of 60 Gbaud and a computational density of 102 TOPS/mm². The main architecture is based on a star coupler with on-chip coupler, limited to 8 delays in the proposed study.

Whereas implementing Next Generation RC on an integrated photonic chip would have several advantages even over conventional optical RC, namely a significantly shorter learning length, and fewer hyperparameters, 1. it is not clearly demonstrated in the article how the proposed architecture offers changeability and reliability and prospects of scalability over other optical RC implementation or even digital NGRC with respect to large-scale chaotic systems.

Response 1

We appreciate the reviewer's questions on changeability, reliability, and scalability. We believe our work indeed presents fundamental advancements in all these aspects. In **Response 2**, we will provide a one-to-one comparison of our work to the classic and recent photonic RC works including those mentioned by the reviewer, to show the advancement of our work. Here is a summary.

Fundamentally, our work is developed from a new reservoir computing framework called the next-generation reservoir, which is distinct from the traditional RC framework that all other current photonic RCs follow. NG-RC eliminates the need for recurrent connections by replacing the recurrent layer with a feedforward network driven by time-delayed inputs. This simplification not only streamlines the system architecture but also surpasses the computational performance of traditional RC systems. Inherent from the good features of the next-generation RC, our system naturally leads to advancement, especially in terms of computational power, compared to other RC systems.

However, our contribution goes beyond merely finding a photonic implementation of the next-generation RC. Instead, we discover a streamlined photonic design that leads to the first realization of next-generation RC on a chip and results in the highest-speed RC system demonstrated to date. Our RC chip employs a passive star coupler with delay-line waveguides to replace the reservoir layer in conventional photonic RC systems that typically rely on complex components (those listed in **Table R2.1 in Response 2**). **Through such a streamlined design, this star coupler, together with a regular optical signaling system, is capable of achieving best-in-class computational capacity, as demonstrated by various intelligent tasks presented in the paper** including various tasks done by other works shown in **Table R2.1** and two sequence prediction tasks (Santa Fe Laser task and Lorenz63 task), two system

emulation tasks (Nonlinear Channel Equalization task and NARMA10 task), and a classification task (COVID-19 task). Importantly, it is this streamlined design that **overcomes the speed and bandwidth limitations** typically seen in photonic RC systems and enables the system to achieve **the highest speed and computing density**. **And this streamlined design also ensures reliability and scalability advancement**, as well as **simplifies the requirement on changeability**, when compared to other photonic RC and optical neural network systems.

Changeability: For an RC, an advantage compared to a conventional neural network is that the reservoir layer can be random and does not need to be changed (or programmed). Only the readout layer requires optimization. The readout layer ensures the changeability of the overall system.

In our experiment, similar to almost all photonic RC systems [1-6,8-11], we only implement the output layer digitally and do not demonstrate it on the photonic chip, as it is not the core novelty of this work. However, the output layer, which is a simple regression layer, can be conveniently realized by programmable MZI arrays, PIN attenuators or MRR weight banks (please see **Figure 1** in the revised manuscript).

The reviewer also questioned the changeability of the nonlinear functions. The nonlinear function is a quadratic function realized by the PD. This may appear a common way to realize optoelectrical nonlinearity. **However, we would like to emphasize that, for next-generation RC, such a quadratic function is already the optimal nonlinear function, and therefore does not need optimization (i.e., changeability).** This is very different from other works; other works use PD to generate nonlinearity because it is convenient, but such nonlinearity is not the best choice for them (most other systems prefer using a sigmoid function that is hard to achieve in optical systems). In contrast, the quadratic function is already optimal for next-generation RC. **In short, although the nonlinear functions in our system are not changeable or programmable, they are already optimal in our computing framework and thus do not require changeability.**

Reliability: We also believe that our work offers significantly higher reliability compared to existing approaches. This reliability is rooted in our streamlined design, which uses a star coupler as the core device. **Importantly, our original manuscript mentioned—but did not emphasize enough—that our system can function optimally even with significant fabrication variances in the star coupler.** This is because it operates effectively with any star coupler, provided it represents an arbitrary full-rank matrix. This inherent simplicity ensures optimal performance within the next-generation RC framework, thereby providing high reliability and error tolerance for our RC system.

In stark contrast, other photonic RC systems rely on much more complicated setups and devices (as shown in **Table R2.1 in Response 2**), which inherently pose greater challenges to reliability. These systems also require delicate optimization of meta-parameters, such as the optimal input ports and delay line distances in [1,2,3], which are often not optimizable for optical systems or are extremely difficult to identify and stabilize for optimal operations, such as those based on laser injections [8]. This complexity introduces significant reliability issues that our streamlined design successfully avoids.

Scalability: Our system is highly scalable in the spatial, time, and wavelength domains.

From a computational theory perspective, conventional RC uses randomly sampled matrices to define the underlying recurrent neural network. These matrices are typically not optimized. When translating such an RC algorithm into a physical system, this non-optimized configuration results in a clumsy system with unnecessary devices, leading to low computing density and difficulty in scaling up. In contrast, a major advancement of the next-generation RC is that it finds optimal matrices, thus optimizing and minimizing the number of required devices and output ports. **Therefore, we can realize the core function of next-generation RC (equivalent to the reservoir layer in conventional RC) using a simple star coupler with a size of only 0.04 mm². This makes our system highly scalable in the spatial domain.**

We compare the chip area per node to other photonic RC systems, where the node number determines the computational capacity. Our calculations include the input ports and the reservoir layer but exclude the output layer, as it is a standard linear regression layer and most of work is done offline on a computer. As shown in **Table R2.1**, **the required chip area per node in our work is the minimum (Also note that our work already requires the minimum number of output ports). Increasing the node number only requires a very slight increase in the size of the star coupler to accommodate more input and output waveguides, without changing its design or adding additional components.** We estimate that a star coupler with an area of 6.25 mm² can support more than 5000 nodes, and this number is only limited by the optical power to ensure a reasonable signal-to-noise ratio.

The number of computing nodes can be further significantly increased through wavelength-division multiplexing, without the need to expand the area or increase the output port number of the star coupler. In the revised manuscript, we have added a new experiment that demonstrates the scaling of the computing power by encoding inputs at parallel wavelengths, with results illustrated in **Figure 5** in the revised manuscript. By leveraging wavelength-division multiplexing and an optical comb source, the effective number of computing nodes can be amplified hundreds of times, with the exact increase depending on the number of available wavelengths.

Furthermore, since our system has the advantage of ultra-high speed, it can also conveniently scale up with time-division multiplexing. We evaluate the number of effective nodes in 1 cm² and 1 μs of various schemes (see **Table R2.1** in **Response 2**). The results show that the number of nodes, which determines the computational capacity, is up to four orders of magnitude higher than that of other schemes.

Table R2.1 compares the scalability of our system with various prior works, and we hope these results will convince the reviewer of the significant scalability advantages of our approach, together with other advantages in footprint, operation speed, energy consumption, and task performance.

Revision 1

1. To better emphasize the novelty and strengths of our work, we rewrite the introduction and add a new Table in Supplementary Note 9 (i.e., **Table R2.1**) to provide a detailed performance comparison of our work with others. We hope these revisions can clearly highlight the significant advantages of our approach in terms of computing performance and high processing speed with minimal resources (chip area, power consumption, output port number).

2. To support our scalability claim using WDM, we add new experimental results and their working principle. These results, illustrated in **Figure 5** of the revised manuscript, demonstrate the scaling of computing power by encoding inputs at parallel wavelengths.

3. In response to the changeability concern and to showcase the adaptability of our system to different tasks in response, we add two experimental demonstrations beyond the three performed in the original manuscript: a sequence prediction task and a nonlinear channel equalization task, with results shown in **Figures 3** and **4** in revised manuscript, together with simulation demonstrations on a variety of tasks done by other RC works as shown in **Table R2.1**. By further comparing our system’s performance to others on these tasks, we highlight its adaptability and emphasize its advantages in computing performance with minimal resource requirements.

Comment 2

This paper might be highly interesting and even it presents experimental results with high-speed, it does not demonstrate the advantages of the proposed passive architecture in comparison with other optical NGRC and optical conventional RC regarding the task treated (see ref [20-28] and also arXiv [35,36])

Response 2

To further illustrate the advancement of our system in terms of computing performance with minimal resources (chip area, power consumption, output port number), we provide a detailed comparison with classic photonic RC systems, **especially those listed by the reviewer**. This comparison covers components, operation speed, scalability, and task performance. The table demonstrates that our system achieves the highest processing speed and computing performance while using minimal resources, including chip area, power consumption, and output port number.

Table R2.1 Performance comparison with previous photonic reservoir computing systems. When achieving the same results as the reference works, the required node count, corresponding chip area, and energy consumption in our work are highlighted in **red**.

Systems	Components required for reservoir layer	Operation Speed ¹ (GHz)	Chip Area Per Node ²	Effective Node No. Per Area in Unit Time ³	Task Performance ⁴		
					Task	Node Count ⁵	Chip Area ⁶
Our work							
 (our work)	On-chip delay lines, star coupler.	60	0.04 mm²	4.7×10⁸ (/1 cm²/1 μ s)	See details below Other work /Our work		
Conventional RC							

[REDACTED] [1]	On chip delay lines and MMIs.	0.125~12.5	1 mm ²	1.3×10 ⁶ (/1 cm ² /1 μ s)	XOR BER = 0	11 / 6	16 mm ² /0.32 mm²	1.7 nJ (digital readout) 1.4 nJ (optical readout) /53 pJ (optical readout)
[REDACTED] [2]	Fiber delay line, couplers, attenuator, piezoelectric fiber stretcher,	9×10 ⁻⁴	N/A	45 (/1 μ s)	NARMA10 NMSE = 0.046	300 /78	N/A /1.55 mm²	1.3 uJ (digital readout) 1.4 uJ (optical readout) /113 pJ (optical readout)
[REDACTED] [3]	Phase modulator, circulator, fiber delay line, PD, low pass filter, RF driver.	1.6×10 ⁻²	N/A	5.9×10 ³ (/1 μ s)	Spoken digital recognition TI46 WER ≈ 0	371 /378	N/A /5.7 mm²	87 nJ (digital readout) 87 nJ (optical readout) /364 pJ (optical readout)
[REDACTED] [4]	Scattering medium, collimator lens system, expander lens system	4×10 ⁻⁹	N/A	0.33 (/1 cm ² /1 μ s)	Kuramoto-Sivashinsky time series NRMSE = 0.298	50000 /2500	60 mm ² /33 mm²	3.6 J (digital readout) /2.1 nJ (optical readout)
[REDACTED] [5]	On-chip multimode waveguide	12.5	0.06 mm ²	8.1×10 ⁷ (/1 cm ² /1 μ s)	Santa Fe NMSE = 0.039	270 /28	4 mm ² /0.75 mm²	4.5 uJ (digital readout) /71 pJ (optical readout)
[REDACTED] [6]	MZI mesh, on-chip delay lines, coherent cavities, variable optical attenuator,	1.9	41 mm ²	7.3×10 ⁴ (/1 cm ² /1 μ s)	Santa Fe NMSE = 0.06	256 /15	658 mm ² /0.51 mm²	6.8 nJ (digital readout) 3.7 nJ (optical readout) /60 pJ (optical readout)

[REDACTED] [7]	EDFA, phase modulator, RF source, RF amplifier, fiber delay lines, fiber couplers, programmable spectral filter, PD, MZM modulator	0.01	N/A	2.8×10^2 (/1 μ s)	nonlinear channel equation (24 dB) SER = 10^{-4}	40 /45	N/A /1 mm²	350 nJ (optical readout) /85 pJ (optical readout)
[REDACTED] [8]	Fiber couplers, circulators, fiber delay lines, attenuators, slave lasers, EDFAs, polarization controllers	0.25	N/A	8×10^{-4} (/1 μ s)	25 Gbps OOK 50km BER = 10^{-3}	240 /105	N/A /1.95 mm²	14 nJ (digital readout) 14 nJ (optical readout) /135 pJ (optical readout)
[REDACTED] [9]	Collimator lens system and expander lens system	2×10^{-9}	N/A	0.054 (/1 cm ² /1 μ s)	N/A			
Other photonic NG-RC								
[REDACTED] [10]	ground glass diffuser, collimator lens system, expander lens system	1×10^{-8}	N/A	7.3×10^{-4} (/1 cm ² /1 μ s)	Kuramoto-Sivashinsky time series NRMSE = 0.298	2500	380 mm ² /33 mm²	1.45 J /2.13 nJ (optical readout)
[REDACTED] [11]	Amplitude modulator, fiber circulator,	1.7×10^{-4}	N/A	1×10^3 (/1 μ s)	Lorenz 63 NRMSEy = 1.23×10^{-2} NRMSEz = 1.89×10^{-2}	1000	N/A /13.9 mm²	9.5 μ J (digital readout) 10.4 μ J (optical readout) /883 pJ (optical readout)

¹Operation speed means the prediction, emulation, or classification speed in photonic reservoir computing.

²Chip area only includes the area of input ports and the reservoir layer area but excludes the output layer, as it is a standard linear regression layer and most of work is done offline on a computer.

³Number of reservoir computing nodes achievable within a 1 square centimeter space and a 1 microsecond time frame.

⁴For a specific task, our system achieves the same simulation results as the reference works. The parameters of our optical engine are highlighted in red.

⁵Number of reservoir computing nodes required to complete a specific task.

⁶Area required for reservoir computing to complete a specific task.

⁷Theoretical energy consumed to perform a single prediction or classification.

We hope this comparison will clearly demonstrate that our proposed system shows significant advancements in computing performance, adaptability, reliability, and scalability, in addition to its significant speed advantage.

In addition to the comparison to the photonic systems, we also compare our system with the high-end GPU H100 in terms of energy efficiency scaling with the node number. The results are shown in **Figure R2.2**. Our optical engine is composed of the following key components: a PPCL500 laser (1 W), an EDFA (0.35 W)[12], a 240 GSamples/s DAC and an on-chip modulator (about 0.75 W)[13,14], ninety metal heaters for the output layer (approximately 2.25 W), and two on-chip photodetectors and a 256 GSamples/s ADC (about 0.75 W)[15]. Therefore, the optical engine, **operating at a line rate of 60 GHz** (the same as our experiment) and **performing 211 Tera operations per second**, has an energy efficiency of around 41 TOPS/W (see Supplementary Note 4). In contrast, NVIDIA's H100 achieves an energy efficiency of roughly 0.15 TOPS/W cited from [16]. Significantly, as shown in **Figure R2.2**, when the reservoir size further increases, our optical engine offers an even more substantial energy consumption advantage over GPU H100 when processing large-scale NG-RC.

Figure R2.2 The energy consumption per operation of NG-RC with different number of outputs completed by our optical engine and GPU H100.

Revision 2

1. We add **Table R2.1** in Supplementary Note 9, demonstrating that our proposed system shows significant advancements in computing performance, adaptability, reliability, and scalability, in addition to its significant speed advantage.

2. We add the energy consumption of the system, including the optical readout layer and peripheral components such as the laser, modulator, EDFA, PD, and ADC, to the Discussion section (lines 373-383) and Supplementary Note 4. A comparison between our optical engine and the GPU H100 is also included in Supplementary Note 4.

Comment 3

In fact, because of the passive nature of the star coupler, it is not possible to adjust critical parameters such as the weight and the non-linear activation function, which constitute a clear limitation to the proposed method to solve more complex tasks.

Response 3

Thank you for the question regarding the programmability of our system. The programmability of an RC system comes from the readout layer, not the reservoir layer. As we have explained under "changeability" in **Response 1**, the reservoir layer conducted by the star coupler does not need to be programmed. Only the weights in the readout layer require programmability, which can be conveniently achieved using programmable MZI arrays, PIN attenuators, or MRR weight banks. The readout layer ensures the optimization of the overall system. In the revised manuscript, we provide the design of the optical readout layer using programmable MRR weight banks in **Figure 1**.

Regarding the adjustability of the nonlinear functions, as we have explained in **Response 1**, the quadratic function realized by the PD is already the optimal nonlinear function and thus does not require adjustability. This is very different from other works, where PD is used to generate nonlinearity because it is convenient, but such nonlinearity is not the best choice for them and requires optimization. In short, although the nonlinear functions in our system are not adjustable or programmable, they are already optimal within our computing framework and thus do not require adjustment.

Revision 3

1. **Figure 1** in our manuscript is revised to show the design chip including the optical readout layer using a programmable MRR array. The working principle is added in lines 147-167.
2. In response to concerns about changeability and to demonstrate the adaptability of our system for solving complex tasks, we have added two additional experimental demonstrations, beyond the three included in the original manuscript: a sequence prediction task and a nonlinear channel equalization task, with the results presented in **Figures 3** and **4** in revised manuscript. Our system outperforms existing works on these tasks, demonstrating its adaptability and computational performance while maintaining minimal resource requirements.

Comment 4

This is the major criticism here, of course, there is an integrated device that can process with high-speed, but only part of it is on chip and is passive (what about the EO modulator, EDFA, laser and Photodiode...).

Response 4

The main intellectual contribution of our work is the development of a streamlined photonic design to realize the reservoir layer for next-generation on-chip reservoir computing (RC) systems, achieving the highest-speed RC system demonstrated to date. Our focus is on the reservoir layer, as our design effectively replaces the complex components traditionally used in

conventional photonic RC systems—often with significant speed limitations—with a simple, passive star coupler and delay-line waveguides. A detailed comparison can be found in **Table R2.1**.

Other optical devices, such as the electro-optic (EO) modulator and photodiode, are standard for optical signaling and detection. Their integration is straightforward and can be readily accomplished by commercial foundries. As is common in the optical neural network field [17,18,19], we do not integrate these optical signaling and detection devices (e.g., EO modulators and photodiodes) onto the chip. However, these components are widely available through silicon photonic foundries and can be easily integrated, with no additional technological advancements needed to match the performance we achieved using off-the-shelf devices. Similarly, the laser and erbium-doped fiber amplifier (EDFA) can be integrated using standard chip coupling methods.

We agree with the reviewers that discussing the prospects for full integration is essential. In our revised manuscript, we provide an outlook on the feasibility of full integration, particularly of the readout layer, as shown in **Figure 1**. Adding an optical output layer would require only a single optical output digitizer. Additionally, we project the hardware performance, including energy efficiency, of the all-optical engine and compare it with the GPU H100. The results are shown in **Figure R2.2**, with detailed calculations provided in **Response 2**.

Revision 4

1. **Figure 1** in our manuscript is revised to show the design chip including the optical readout layer using the MRR array with its working principle of the optical readout layer between lines 147 and 167.
2. The discussion of full integration is added between lines 358 and 372.
3. We estimate the energy consumption of the system, including the optical readout layer and peripheral components such as the laser, modulator, EDFA, PD, and ADC. Details are provided in the Discussion section (lines 373–383) and Supplementary Note 4. A comparison between our optical engine and the GPU H100 is also included in Supplementary Note 4.

Reference

1. Vandoorne, Kristof, et al. "Experimental demonstration of reservoir computing on a silicon photonics chip." *Nature communications* 5.1 (2014): 3541.
2. Vinckier, Quentin, et al. "High-performance photonic reservoir computer based on a coherently driven passive cavity." *Optica* 2.5 (2015): 438-446.
3. Larger, Laurent, et al. "High-speed photonic reservoir computing using a time-delay-based architecture: Million words per second classification." *Physical Review X* 7.1 (2017): 011015.
4. Rafayelyan, Mushegh, et al. "Large-scale optical reservoir computing for spatiotemporal chaotic systems prediction." *Physical Review X* 10.4 (2020): 041037.
5. Sunada, Satoshi, and Atsushi Uchida. "Photonic neural field on a silicon chip: large-scale, high-speed neuro-inspired computing and sensing." *Optica* 8.11 (2021): 1388-1396.
6. Nakajima, Mitsumasa, Kenji Tanaka, and Toshikazu Hashimoto. "Scalable reservoir

- computing on coherent linear photonic processor." *Communications Physics* 4.1 (2021): 20.
7. Lupo, Alessandro, et al. "Deep photonic reservoir computer based on frequency multiplexing with fully analog connection between layers." *Optica* 10.11 (2023): 1478-1485.
 8. Shen, Yi-Wei, et al. "Deep photonic reservoir computing recurrent network." *Optica* 10.12 (2023): 1745-1751.
 9. Antonik, Piotr, et al. "Human action recognition with a large-scale brain-inspired photonic computer." *Nature Machine Intelligence* 1.11 (2019): 530-537.
 10. Wang, Hao, et al. "Optical next generation reservoir computing." *arXiv preprint arXiv:2404.07857* (2024).
 11. Cox, Nicholas, et al. "Photonic next-generation reservoir computer based on distributed feedback in optical fiber." *arXiv preprint arXiv:2404.07116* (2024).
 12. Feng, Hanke, et al. "Integrated lithium niobate microwave photonic processing engine." *Nature* 627.8002 (2024): 80-87.
 13. Kossel, Marcel A., et al. "8.3 An 8b DAC-based SST TX using metal gate resistors with 1.4 pJ/b efficiency at 112Gb/s PAM-4 and 8-Tap FFE in 7nm CMOS." 2021 IEEE International Solid-State Circuits Conference (ISSCC). Vol. 64. IEEE, 2021
 14. Wang, Cheng, et al. "Integrated lithium niobate electro-optic modulators operating at CMOS-compatible voltages." *Nature* 562.7725 (2018): 101-104.
 15. Khairi, Ahmad, et al. "A 1.41-pJ/b 224-Gb/s PAM4 6-bit ADC-based SerDes receiver with hybrid AFE capable of supporting long reach channels." *IEEE Journal of Solid-State Circuits* 58.1 (2022): 8-18.
 16. Xu, Zhihao, et al. "Large-scale photonic chiplet Taichi empowers 160-TOPS/W artificial general intelligence." *Science* 384.6692 (2024): 202-209.
 17. Feldmann, Johannes, et al. "Parallel convolutional processing using an integrated photonic tensor core." *Nature* 589.7840 (2021): 52-58.
 18. Dong, Bowei, et al. "Higher-dimensional processing using a photonic tensor core with continuous-time data." *Nature Photonics* 17.12 (2023): 1080-1088.
 19. Bai, Bowen, et al. "Microcomb-based integrated photonic processing unit." *Nature Communications* 14.1 (2023): 66.

Response To Reviewer 3 (R3)

Comment 1

Report on A 103-TOPS/mm² Integrated Photonic by D. Liang et al.

In this paper the authors report an integrated optics implementation of a “Next Generation Reservoir Computer”. The paper is innovative, and will interest the community working on photonic neuromorphic computing. The integrated optics system reported seems rather standard. The tasks addressed are not amongst the most challenging. On the other hand the speed achieved is impressive. I have a number of substantial remarks that make it difficult to give a definitive judgement on the paper. I would need to see a revised version to give a definitive opinion, although I tend to think it will be a bit below the level sought for in Nature Communications.

Response 1

We thank the reviewer for being willing to give us the chance to further clarify the significance and advancement of this work.

Regarding the comment that 'the integrated optics system reported seems rather standard,' we would like to clarify that while our system may appear standard to the photonics community, **it effectively replaces the entire input and reservoir layers in traditional photonic RC systems, which typically require more complex components and face significant speed limitations.** Moreover, our streamlined design achieves **best-in-class computational power**, even when compared to other reservoir computing systems that comprise much more complicated architectures. This is in addition to its **huge speed advantage.**

To show this point, we make a detailed one-by-one comparison with many photonic RC systems. We evaluate the computational performance of different systems on the same tasks and assess the required resources, such as power consumption, chip footprint, and number of output nodes. The results are summarized in **Table R3.1**. This table highlights that our system offers significant advantages in computing performance, adaptability, reliability, and scalability, in addition to its notable speed advantage, compared to other photonic RC systems demonstrated to date. Moreover, this comparison addresses the concern that "the tasks addressed are not among the most challenging." By performing a variety of tasks performed by other systems in simulations [1-11], we show that our system consistently matches their performances across all tasks while using minimal resources, particularly in terms of the number of output nodes. Beyond the simulations summarized in **Table R3.1**, we also present two additional experimental demonstrations—sequence prediction and nonlinear channel equalization—beyond the three experiments described in the original manuscript. The results, shown in the **new Figures 3 and 4** in our manuscript, further highlight our system’s superior computing performance in comparison to others.

We want to emphasize **these advantages come from the two key intelliential contributions and novelties** from our work.

First, our work is developed from a new reservoir computing framework called the next-

generation reservoir, which is distinct from the traditional RC framework that all other current photonic RCs follow. NG-RC eliminates the need for recurrent connections by replacing the recurrent layer with a feedforward network driven by time-delayed inputs. This simplification not only streamlines the system architecture but also surpasses the computational performance of traditional RC systems. Inherent from the good features of the next-generation RC, our system naturally leads to advancement, especially in terms of computational power, compared to other RC systems.

Second, we discover a streamlined photonic design that leads to the first realization of next-generation RC on a chip and results in the highest-speed RC system demonstrated to date. Our RC chip employs a passive star coupler with delay-line waveguides to replace the reservoir layer in conventional photonic RC systems that typically rely on complex components (those shown in **Table R3.1**). Through such a streamlined design, we overcome the speed and bandwidth limitations typically seen in photonic RC systems and enable the system to achieve **the highest speed and computing density. And this streamlined design also ensures reliability and scalability advancement, as well as simplifies the requirement on changeability**, when compared to other photonic RCs and optical neural network systems.

Therefore, the “standard appearance” of our system should not be the reason to undermine its significance.

Table R3.1 Performance comparison with previous photonic reservoir computing systems. When achieving the same results as the reference works, the required node count, corresponding chip area and energy consumption in our work are highlighted in red.

Systems	Components required for reservoir layer	Operation Speed ¹ (GHz)	Chip Area Per Node ²	Effective Node No. Per Area in Unit Time ³	Task Performance ⁴			
					Task	Node Count ⁵	Chip Area ⁶	Energy Consumption ⁷
Our work								
 (our work)	On-chip delay lines, star coupler.	60	0.04 mm²	4.7×10⁸ (/1 cm²/1 μs)	See details below Other work /Our work			
Conventional RC								
[REDACTED] [1]	On chip delay lines and MMIs.	0.125~12.5	1 mm ²	1.3×10 ⁶ (/1 cm ² /1 μs)	XOR BER = 0	11 / 6	16 mm ² /0.32 mm²	1.7 nJ (digital readout) 1.4 nJ (optical readout) /53 pJ (optical readout)

[REDACTED] [2]	Fiber delay line, couplers, attenuator, piezoelectric fiber stretcher, polarization controller.	9×10^{-4}	N/A	45 (/1 μ s)	NARMA10 NMSE = 0.046	300 /78	N/A /1.55 mm²	1.3 uJ (digital readout) 1.4 uJ (optical readout) /113 pJ (optical readout)
[REDACTED] [3]	Phase modulator, circulator, fiber delay line, PD, low pass filter, RF driver.	1.6×10^{-2}	N/A	5.9×10^3 (/1 μ s)	Spoken digital recognition TI46 WER \approx 0	371 /378	N/A /5.7 mm²	87 nJ (digital readout) 87 nJ (optical readout) /364 pJ (optical readout)
[REDACTED] [4]	Scattering medium, collimator lens system, expander lens system	4×10^{-9}	NA	0.33 (/1 cm ² /1 μ s)	Kuramoto-Sivashinsky time series NRMSE = 0.298	50000 /2500	60 mm ² /33 mm²	3.6 J (digital readout) /2.1 nJ (optical readout)
[REDACTED] [5]	On-chip multimode waveguide	12.5	0.06 mm ²	8.1×10^7 (/1 cm ² /1 μ s)	Santa Fe NMSE = 0.039	270 /28	4 mm ² /0.75 mm²	4.5 uJ (digital readout) /71 pJ (optical readout)
[REDACTED] [6]	MZI mesh, on-chip delay lines, coherent cavities, variable optical attenuator,	1.9	41 mm ²	7.3×10^4 (/1 cm ² /1 μ s)	Santa Fe NMSE = 0.06	256 /15	658 mm ² /0.51 mm²	6.8 nJ (digital readout) 3.7 nJ (optical readout) /60 pJ (optical readout)
[REDACTED] [7]	EDFA, phase modulator, RF source, RF amplifier, fiber delay lines, fiber couplers, programmable spectral filter, PD, MZM modulator	0.01	N/A	2.8×10^2 (/1 μ s)	nonlinear channel equation (24 dB) SER = 10^{-4}	40 /45	N/A /1 mm²	350 nJ (optical readout) /85 pJ (optical readout)

[REDACTED] [8]	Fiber couplers, circulators, fiber delay lines, attenuators, slave lasers, EDFAs, polarization controllers	0.25	N/A	8×10^{-4} (/1 μ s)	25 Gbps OOK 50km BER = 10^{-3}	240 /105	N/A /1.95 mm²	14 nJ (digital readout) 14 nJ (optical readout) /135 pJ (optical readout)
[REDACTED] [9]	Collimator lens system and expander lens system	2×10^{-9}	NA	0.054 (/1 cm ² /1 μ s)	N/A			
Other photonic NG-RC								
[REDACTED] [10]	ground glass diffuser, collimator lens system, expander lens system	1×10^{-8}	NA	7.3×10^{-4} (/1 cm ² /1 μ s)	Kuramoto-Sivashinsky time series NRMSE = 0.298	2500	380 mm ² /33 mm²	1.45 J /2.13 nJ (optical readout)
[REDACTED] [11]	Amplitude modulator, fiber circulator,	1.7×10^{-4}	N/A	1×10^3 (/1 μ s)	Lorenz 63 NRMSEy = 1.23×10^{-2} NRMSEz = 1.89×10^{-2}	1000	N/A /13.9 mm²	9.5 μ J (digital readout) 10.4 μ J (optical readout) /883 pJ (optical readout)

¹Operation speed means the prediction, emulation, or classification speed in photonic reservoir computing.

²Chip area only includes the area of input ports and the reservoir layer area but excludes the output layer, as it is a standard linear regression layer and most of work is done offline on a computer.

³Number of reservoir computing nodes achievable within a 1 square centimeter space and a 1 microsecond time frame.

⁴For a specific task, our system achieves the same simulation results with the reference works. The parameters of our optical engine are highlighted in red.

⁵Number of reservoir computing nodes required to complete a specific task.

⁶Area required for reservoir computing to complete a specific task.

⁷Theoretical energy consumed to perform a single prediction or classification.

Revision 1

1. In response to concerns regarding the advancements of our work beyond speed, we have made two key additions to the revised manuscript. First, we include **Table R3.1** in Supplementary Note 9, which demonstrates that our proposed system exhibits significant improvements in computing performance, adaptability, reliability, and scalability, alongside its notable speed advantage. Second, we present two additional experimental demonstrations—specifically, a sequence prediction task

and a nonlinear channel equalization task—beyond the three originally performed. The results, shown in the revised **Figures 3** and **4**, further emphasize our system’s superior computing performance compared to other systems.

2. To better emphasize the novelty and strengths of our work, we rewrite the introduction between lines 65 and 95.

To address these challenges, we propose and experimentally demonstrate a novel integrated photonic RC chip based on a new framework called next-generation RC (NG-RC). NG-RC eliminates the need for recurrent connections by replacing the recurrent layer with a feedforward network driven by time-delayed inputs. This simplification not only streamlines the system architecture but also matches, and in some cases surpasses, the computational performance of traditional RC systems. Some other works have also discussed that introducing time-delay inputs to the RC system can improve photonic RC performance. However, these systems still rely on complex recurrent connections in the reservoir layer, which limit scalability and processing speed. Here, building on NG-RC framework, we design a high-speed, ultra-compact photonic RC system integrated on a silicon chip. Our RC chip employs a passive star coupler with delay-line waveguides to implement the feedforward network, effectively replacing the reservoir layer in conventional photonic RC systems that typically rely on complex components. This design enables integrating the reservoir layer on a silicon photonic chip with a footprint of just 2 mm², while still achieving best-in-class computational performance across various benchmarks and practical tasks, compared to much bulkier photonic systems. Since our chip is built using linear optical devices, it overcomes the speed and bandwidth limitations typically seen in photonic RC systems. As a result, we achieve the fastest operation speed to date, with experimentally demonstrated information processing rates exceeding 60 GHz. This speed can be further enhanced with the use of higher-speed optical modulators and photodetectors.

In addition, our design features high scalability, energy efficiency, and remarkable tolerance to fabrication errors, making it well-suited for large-scale computing systems. The system is capable of supporting over 5,000 output nodes, where the number of nodes directly correlates with computational capacity. We further demonstrate that this capacity can be further enhanced through wavelength multiplexing. Notably, NG-RC, with the same number of nonlinear nodes, typically outperforms many traditional RC systems in terms of computational performance. Moreover, the required chip area per nonlinear node and energy consumption in our design are far superior to those of other photonic and electronic systems, including state-of-the-art platforms like the GPU H100. This combination of high performance, high processing speed, compact design, and energy efficiency positions our integrated photonic RC chip as a strong candidate for next-generation computing.

We hope these revisions can clearly highlight the significant advantages of our approach in terms of computing performance and high speed with minimal resources (chip area, power consumption, output port number)

Comment 2

I have some serious reservations about the title and some claims in the paper. Indeed the

authors implement off chip significant parts of their information processing system. The most challenging part, from a photonics point of view, would be to implement the photodetection on chip and to put weights optically or electronically on chip. Doing so could considerably complicate the system. In addition the authors outsource much of the computation using a high end oscilloscope to measure the signals and then digital postprocessing. Overall the authors need to do a fairer evaluation of the performance of their system. What would it look like if the detection part was done on chip? Given the overhead of converting electrical signals to optical, and then optical to electrical followed by digitization, how would the system compare to a fully electronic system? For these reasons the title, the “computing density” discussed in the paper (see e.g.the abstract), and Table 1, all seem to me much too optimistic. Overall I feel that Table 1 is quite misleading, omitting some important papers, and counting energy consumption for instance in a very weird way.

Response 2

The reviewer correctly points out that we did not integrate the readout layer and detection devices onto the photonic chip. This is similar to most works in the related field of photonic RC [1-6,8-14]. However, the readout layer, which is a regression layer, can be conveniently realized using programmable MZI arrays, PIN attenuators, or microring (MRR) weight banks. These components are widely available in silicon photonic foundries and can be easily integrated.

To realize an optical readout layer, one approach is to pass the outputs from the star coupler through an array of programmable MRRs (the first column of MRRs in **Figure 1c** in the revised manuscript). The resonance wavelengths of the MRRs are aligned with the input wavelength. By adjusting the resonance of each MRR using an embedded thermal phase shifter, the fraction of light directed to the Drop ports can be finely controlled. In addition to tuning the signal's amplitude via the phase shifter on the MRR, a second phase shifter on the bus waveguide adjusts the signal phase. The combined tuning of these two phase shifters enables precise control of complex weights. The weighted signals are then detected by a balanced photodetector (BPD). The outputs from the first $M/2$ rows of MRRs are combined and detected by one photodiode, while the remaining MRR outputs are combined and detected by a second photodiode. The differential signal between the two photodiodes provides the full range of weights, including both positive and negative values. The final signal after the BPD is given by

$$\begin{aligned}
 y_{BPD,\lambda} &= |\omega_{MRR}^+ \mathbf{y}_{star,\lambda}^+|^2 - |\omega_{MRR}^- \mathbf{y}_{star,\lambda}^-|^2 \\
 &= \underbrace{c}_{\text{constant}} + \underbrace{\sum_{n=1}^N \omega_{lin,n} x_{t+1-n}}_{\text{linear terms}} + \underbrace{\sum_{m=1}^N \sum_{n=m}^N \omega_{nonlinear,mn} x_{t+1-m} x_{t+1-n}}_{\text{quadratic terms}}
 \end{aligned}$$

where ω_{MRR}^+ and ω_{MRR}^- are complex weight vectors given by the first $M/2$ and second $M/2$ rows of MRRs, respectively, $\mathbf{y}_{star,\lambda}^+$ and $\mathbf{y}_{star,\lambda}^-$ are the output vectors from the first $M/2$ outputs and second $M/2$ outputs of the star coupler, respectively. Eq. above indicates that the output of the BPD is a regression of the constant, input, and quadratic polynomials of inputs. The regression coefficients c , ω_{lin} , and $\omega_{nonlinear}$ can be programmed from -1 to 1 using the

MRR array. Therefore, the outputs of the optical engine have the equivalent function to the NG-RC.

With optical readout, only a balanced photodetector is required, and high-speed photodetectors are mature devices in silicon photonic foundries. No additional advancements are needed to achieve the same performance as we obtained using off-the-shelf devices. In our revised manuscript, we provide an outlook on full integration including the optical readout layer on the chip shown in **Figure 1**.

Regarding the overhead of electrical-to-optical and optical-to-electrical conversion, our original calculation of energy consumption considered this by including the energy consumption of a PPCL500 laser (1 W), an EDFA (0.35W)[15], a 240 GSamples/s DAC and an on-chip modulator (about 0.75 W)[16,17], ninety metal heaters for the output layer (approximately 2.25 W), and two on-chip photodetectors and a 256 GSamples/s ADC (about 0.75 W)[18]. Therefore, the optical engine, **operating at a line rate of 60 Gbaud** (the same as our experiments) and **performing 211 Tera operations per second**, has an energy efficiency of around 41 TOPS/W (see Supplementary Note 4). In contrast, NVIDIA's H100 achieves an energy efficiency of roughly 0.15 TOPS/W [19]. To provide a clearer picture of the energy advantages, we now plot how the energy scales with the effective computing nodes in the system, compared to GPU H100, as shown in **Figure R3.2**. A clear energy advantage is shown even with OE/EO conversion overhead.

Figure R3.2 The energy consumption of NG-RC with different number of outputs completed by our optical engine and GPU H100

Revision 2

1. We have **deleted Table 1** from the original manuscript to address the reviewers' concerns about imprecise descriptions and replaced it with a new table in Supplementary Note 9 (**Table R3.1**). This new table provides detailed performance comparisons with previous photonic reservoir computing systems, covering key metrics such as speed, computing density, nonlinear node count, footprint, and energy consumption for performing the same tasks.
2. **Figure 1** in our manuscript is revised to show the design chip including the optical readout layer using the MRR array with its working principle of the optical readout layer between lines

147 and 167.

3. The discussion of full integration is added between lines 358 and 372.

4. We estimate the energy consumption of the system, including the optical readout layer and peripheral components such as the laser, modulator, EDFA, PD, and ADC. Details are provided in the Discussion section (lines 373–383) and Supplementary Note 4. A comparison between our optical engine and the GPU H100 is also included in Supplementary Note 4.

Comment 3

Concerning terminology, the name “next generation reservoir computing” is a bit pompous. In fact, an alternative way of looking at the experiment is as an “Extreme Learning Machine” (another pompous name) that is adapted to process a time series by feeding it with several delayed inputs. I recommend the authors connect with this alternative interpretation.

The “next generation reservoir computing” is also closely related to

L. Jaurigue, E. Robertson, J. Wolters, and K. Lüdge, “Reservoir computing with delayed input for fast and easy optimisation,” *Entropy* 23, 1560 (2021).

L. Jaurigue and K. Lüdge, “Reducing reservoir computer hyperparameter dependence by external timescale tailoring,” *Neuromorphic Comput. Eng.* (2024).

in which several delayed inputs are fed into a standard reservoir computer. I recommend the authors connect with these works.

I find the approach of the authors very close, in its principle, to some other optical extreme learning machines, as well as to Ref 6 “11 tops photonic convolutional accelerator.” A discussion of this connection, as well as to other works that implement photonic information processing using linear optics followed by photodetection, is called for. Given the way the authors process the time series, this comparison seems very relevant (even if Ref 6 and some other papers do not deal with time series). In fact the third task the authors consider is not based on a time series, and thus clearly falls into the “extreme learning machine” category mentioned above.

Response 3

We understand the reviewer’s concern regarding the title “next-generation reservoir computing.” However, we did not invent this term or intend to exaggerate our work. The phrase “next-generation reservoir computing” refers to a new RC computing framework reported in *Nature Communications* under the title “Next generation reservoir computing.” Other recent photonic implementations of this framework [10,11] have also adopted this name, although demonstrating much slower speeds.

We appreciate the reviewer’s suggestions and have connected our work to these valuable references in the revised manuscript.

L. Jaurigue and K. Lüdge’s research emphasizes the importance of time-delayed inputs, effectively demonstrating their computational benefits. However, from a photonic hardware

perspective, the requirement for a feedback loop to connect the input and output of nonlinear elements presents significant challenges for on-chip implementation due to excessive waveguide loss, which could also hinder processing speed.

The feedforward architecture with delayed input in our work is indeed similar to extreme learning machines. However, extreme learning machines often lack clear and optimal correlations within the feature vector, thus requiring a large output feature vector (or output weights) to achieve optimal corrections, which in turn leads to much higher hardware costs.

Our computing framework addresses these challenges. First, it transforms the original recurrent architecture into a feedforward one, thereby eliminating the need for feedback loops in optical implementations and removing speed constraints. This has resulted in the highest-speed RC system demonstrated to date. Additionally, our framework optimizes the correlations within the feature vector, enabling a reduction in the number of components without significantly affecting the performances, unlike in extreme learning machines.

Beyond the computing framework, we also discover an optimal photonic design that leads to the first realization of such a framework on a chip, leading to the highest-speed RC system demonstrated to date. Our design simplifies conventional photonic RC systems, which typically require complex components, into a simple star coupler. This straightforward design is key to achieving the highest speed and computing density, and it also enhances reliability and scalability while reducing the complexity associated with changeability, particularly when compared to other photonic RCs and optical neural network systems.

Some prior photonic systems may appear similar to ours, particularly in terms of using linear delayed inputs and photodetection. However, in our system, following the framework of next-generation reservoir computing, the delay line does more than merely create delayed inputs, as is common in other systems. Instead, it simultaneously generates the biases, the delayed inputs, and the optimal nonlinear mixture of inputs. The photodetection process in our system also serves a more sophisticated role than merely providing an optoelectrical nonlinearity. Other systems often use photodetectors to generate nonlinearity because it is convenient, however this nonlinearity is not always the best choice for their applications (many systems prefer using a sigmoid function, which is challenging to implement in optical systems). In contrast, the quadratic function provided by our photodetection is already the optimal nonlinear function in next-generation RC. Due to the novel design, our system offers significant advantages in computing performance, adaptability, reliability, and scalability, in addition to its notable speed advantage, compared to other photonic RC systems demonstrated to date, as shown in **Table R3.1**.

Revision 3

We have added these important references in the revised manuscript.

Comment 4

Supplementary material:” Our experiment has nine input data (including constant) and thirty-six corresponding quadratic polynomials. Thus, the star coupler has forty-five
--

output ports in total.”

Please explain this important consideration in more detail. Why 45? (I think I know, but this needs explaining). I would put this important consideration in the main text in section 2.2.

In fact it would be very interesting to understand how many output ports are really needed for good performance. Could the authors give a plot of performance as a function of the number of output ports that are used? What happens if less than 45 output ports are used? Any idea whether using more than 45 output ports could be beneficial?

Response 4

When our input dimension is fixed at 9, we plot the performance of our system as a function of the number of outputs using the NARMA10 task for demonstration, as shown in **Figure R.3.4.1**. When the number of nodes is fewer than 45 (as observed in the experimental results), we cannot fully adjust the coefficients of the 45 degrees of freedom. As the number of nodes increases, the number of coefficients that we can optimize increases as well, leading to a decrease in NMSE. However, when the number of nodes exceeds 45 (simulation results), we can only adjust the coefficients of the 45 degrees of freedom, resulting in a consistent NMSE.

Figure R3.4.1 NMSE of NARMA10 task with increasing number of outputs, when the input dimension is fixed at 9.

The reason behind the result is, after the star coupler, the output vector $\mathbf{y}_{\text{star},\lambda}$ is expressed as:

$$\mathbf{y}_{\text{star},\lambda} = \mathbf{W}_{\text{star},\lambda} \cdot (\mathbf{C} \oplus \mathbf{X})$$

Here, $\mathbf{W}_{\text{star},\lambda}$ denotes an $M \times (N + 1)$ complex matrix, representing the transfer function of the star coupler at wavelength λ , where N is the number of delayed input copies and M is dimension of the output vector.

The outputs from the star coupler are followed by a readout layer, which can be implemented either digitally or optically. In a digital implementation, each output of the star coupler is detected by a photodetector, producing an output given by:

$$\begin{aligned} y_{PD,i} &= |\mathbf{W}_{\text{star},\lambda,i} \cdot (\mathbf{C} \oplus \mathbf{X})|^2 \\ &= \underbrace{c_i}_{\text{constant}} + \underbrace{\sum_{n=1}^N a_{n,i} x_{t+1-n}}_{\text{linear terms}} + \underbrace{\sum_{m=1}^N \sum_{n=m}^N b_{mn,i} x_{t+1-m} x_{t+1-n}}_{\text{quadratic terms}} \end{aligned}$$

where $y_{PD,i}$ is the output of the photodetector, and $c_i, a_{n,i}$, and $b_{mn,i}$ are constants, which are determined by the transfer function of the star coupler. Eq. above shows that the output of each photodetector is a combination of a constant, linear inputs, and their quadratic polynomials. M outputs form a vector \mathbf{y}_{PD} , where each element is a similar mixture but with different coefficients determined by the transfer function matrix of the star coupler. When the output port number $M \geq \frac{(N+1)(N+2)}{2}$, \mathbf{y}_{PD} can be transformed into the feature vector required by the NG-RC, consisting of constant, linear, and nonlinear components, using a linear matrix. This means there exists a \mathbf{W} such that $\mathbf{W} \cdot \mathbf{y}_{PD} = \mathbf{C} \oplus \mathcal{O}_{\text{linear}} \oplus \mathcal{O}_{\text{nonlinear}}$, where $\mathbf{C} \oplus \mathcal{O}_{\text{linear}} \oplus \mathcal{O}_{\text{nonlinear}}$ is the feature vector in the digital NG-RC. In this case, the system functions equivalently to the NG-RC.

In our chip, the number of input ports is 9, 8 for delayed input copies, and 1 for the constant term. And the output number is $45=(9 \times 10)/2$. Therefore, the feature vectors from the star coupler have the same computational power as the independent elements in the digital NG-RC.

We realize that this point is not clearly explained in our original manuscript. In the revised version, we have now clarified this point.

When we increase the input port number, which is $N+1$ (N signals and a constant), and the number of distinct output ports accordingly changes to $(N+1)(N+2)/2$, we can observe the NMSE decreases exponentially with the number of nodes in the **Figure R3.4.2**. In **Figure R3.4.2**, we also compare our system with other works performing the same task. The results highlight a key advantage of our system: it achieves significantly lower NMSE with fewer feature vectors. This leads to reduced energy consumption and a smaller chip footprint.

Figure R3.4.2 NMSE of NARMA10 task with increasing number of outputs, when the input dimension changes accordingly with the number of output nodes.

Revision 4

1. We add a paragraph to further clarify the feature vector in digital NG-RC, especially its dimension and elements, on lines 114-120.

It is worth noting that in the NG-RC, the dimension of the output vector $\mathcal{O}_{\text{total}}$ is significantly smaller than that in comparable RC systems. When the size of the input vector is $N + 1$, the size of the $\mathcal{O}_{\text{total}}$ only needs to be $(N + 1) \times (N + 2)/2$, which includes a constant C, N inputs, and $N \times (N + 1)/2$ quadratic polynomials of inputs. The resulted y_{out} is the linear

combinations of the elements in O_{total} with weights trained by linear regression.

We further explain why our system has the same computational power to the digital NG-RC on lines 136-143.

Eq.4 shows that the output of each photodetector is a combination of constant, linear inputs, and their quadratic polynomial. M outputs form a vector \mathbf{y}_{PD} , where each element is a similar mixture but with different coefficients determined by the transfer function matrix of the star coupler. When the number of outputs $M \geq (N + 1) \times (N + 2)/2$, \mathbf{y}_{PD} can be transformed into the feature vector required by the NG-RC, consisting of constant, linear, and nonlinear components, using a linear matrix. In this case, the system functions equivalently to the NG-RC.

We also add why our chip has 45 outputs, on lines 199-201.

The star coupler occupies a footprint of 0.04 mm^2 , with 9 inputs (one for unmodulated light and eight for delayed signal copies) and 45 (i.e., $(9+1) \times 9/2$) outputs to generate the necessary feature vector.

Comment 5

What if the task requires more than quadratic polynomials in the input? Could the system be adapted to this case?

Response 5

To handle polynomials beyond quadratic, cascaded modulators can be used at the input stage, as illustrated in **Figure R3.5**. The input includes terms c , x , and x^2 . These signals are combined via a star coupler, weighted by a microring weight bank, and then detected by a balanced photodetector. This setup enables the implementation of third-order and fourth-order terms. To obtain even higher-order terms, additional modulators can be cascaded.

Figure R3.5 Photonic reservoir computing framework for generating higher order of terms.

Revision 5

We have added a discussion on the method for generating nonlinear terms beyond quadratic polynomials in the input on lines 411-419, with a detailed explanation provided in Supplementary Note 8.

Comment 6

Fig. 1 is a bit cramped vertically. I don't understand the text "exact equivalence" in the figure.

Response 6

Thank you for pointing out this issue.

We have redrawn **Figure 1** in our manuscript to better illustrate the concept and principles of our work.

Comment 7

End of section 1. "Metaparamters" : missing E
"and providing interpretable results » not clear what this means.

Response 7

Thank you for spotting the typo. We have corrected it in the revised manuscript.

What we mean by "providing interpretable results" is that NG-RC does not require meta-parameters that demand careful optimization or extensive learning. For instance, it is straightforward to see that increasing the number of input delay lines N improves performance, while the number of outputs is determined by the relationship $(N+1)(N+2)/2$ based on the input.

In contrast, many previous works lack clear guidelines for determining optimal meta-parameters, therefore requiring much more complex learning processes.

Comment 8

Section 2, after Eq. 1 "the quadratic polynomial feature vector has been demonstrated good prediction » garbled.

After Eq. 2 "The neighboring delay line introduces a time delay of Δt ," Probably garbled (check meaning and plurals). There are 8 delay lines.

Response 8

Thank you for pointing out the issue. We have revised the sentences on lines 123-126.

The input is then split into N delayed copies sent into a star coupler together with an unmodulated laser representing the constant C . The neighboring delay line introduces a time delay of Δt , which equals to one symbol duration of the input signal.

Comment 9

End of section 2.1 "Importantly, the actual values of the c_i , $a_{i,k}$, and $b_{i,p,q}$ do not affect the final computing output because the final readout layer can adjust them to the optimal

values through training.” This is wrong, except if n is large enough (and even when n is large, there are values of c, a, b for which there will be a problem). Please check what you mean.

Response 9

Using our system, the i -th output of the star coupler after photodetection is

$$\begin{aligned}
 y_{PD,i} &= |\mathbf{W}_{\text{star},\lambda,i} \cdot (C \oplus \mathbf{x})|^2 \\
 &= \underbrace{c_i}_{\text{constant}} + \underbrace{\sum_{n=1}^N a_{n,i} x_{t+1-n}}_{\text{linear terms}} + \underbrace{\sum_{m=1}^N \sum_{n=m}^N b_{mn,i} x_{t+1-m} x_{t+1-n}}_{\text{quadratic terms}}
 \end{aligned}$$

The equation shows the output is a mixture of constant, linear inputs and quadratic polynomials of inputs.

M outputs form a vector \mathbf{y}_{PD} , where each element is a similar mixture but with different coefficients. These coefficients are determined by the transfer function of the star coupler. When the output port number $M \geq \frac{(N+1)(N+2)}{2}$, \mathbf{y}_{PD} can be transformed into the feature vector required by the NG-RC, consisting of constant, linear, and nonlinear components, using a linear matrix. This means there exists a \mathbf{W} such that $\mathbf{W} \cdot \mathbf{y}_{PD} = C \oplus O_{\text{linear}} \oplus O_{\text{nonlinear}}$, where $C \oplus O_{\text{linear}} \oplus O_{\text{nonlinear}}$ is the feature vector in the digital NG-RC. In this case, the system functions equivalently to the NG-RC. Although different coefficients $c_i, a_{n,i}, b_{mn,i}$ lead to different matrices \mathbf{W} , we can always determine the optimal weights for the readout layer through training, ensuring performance that is exact to that of the digital NG-RC.

Revision 9

We revise the principle of our system to clarify why our system has the equivalent function of NG-RC (lines 136-146).

Comment 10

Section 2.3.1 Lorenz system.

Eq. 6. Please specify the time step you use, as this defines the complexity of the task.

Fig 3b: I don't understand why this picture is informative. Indeed, for one step ahead prediction, except if the prediction is very bad, one will reproduce approximately the Lorenz attractor. Please clarify or remove.

Fig 3c and d: why is the horizontal axis in ns? There are no time units in Eq. 6 (see also remark about the time step used). The horizontal axis is probably the real time used to input the data, but this is not explained.

Below eq. 7, the NMSE for the Lorenz task is reported. Please compare with what has been achieved previously in the literature.

Response 10

For the Lorenz63 task, we use a time step of 0.05 when numerically solving the equation to generate the training and testing datasets. The horizontal axis is in ns because our system completes each point prediction in 16.7 picoseconds.

In this task, what we do is not a simple one-step-ahead prediction like most works do [23,24]. Instead, we predict the z-axis information using the x and y-axis data from the Lorenz system. This cross-prediction is significantly more challenging than a one-step-ahead prediction because we lack prior information about the z-axis. If the prediction is poor, the resulting z-axis waveform will be completely distorted, and we will not be able to approximate the Lorenz attractor.

This cross-prediction task is also done in [25], where an NMSE reported 0.009 but using 2000 output nodes. In contrast, our experiment achieves an NMSE of 0.0143 with only 45 nodes. This NMSE can be further reduced by employing a modulator with a higher bandwidth (currently, we use a 40 GHz modulator to generate a 60 Gbaud signal). When simulating the same operation digitally, without the constraints of bandwidth limitations and noise, the NMSE can be reduced to 3×10^{-4} with 45 nodes.

In addition, compared to the system in [25], our optical engine significantly outperforms in terms of speed and energy efficiency. Specifically, the system in [25], requires 0.1 seconds for each prediction and consumes about 1.45 J of energy per prediction. In comparison, our projected optical engine with optical weighting performs each prediction in just 16.7 ps and consumes only 85 pJ. This demonstrates that our optical engine offers superior computational speed and energy efficiency.

Revision 10

We add some context about the comparison about the NMSE of the Lorenz task on lines 277-282.

The experimental NMSE for the Lorenz63 task is 1.43×10^{-2} using only 45 output nodes. This NMSE can be further reduced by utilizing a modulator with a higher bandwidth (currently, a 40 GHz modulator generates a 60 Gbaud signal). In digital simulations, without bandwidth constraints, the NMSE can be reduced to 3×10^{-4} with 45 nodes. In comparison, [35] reports an NMSE of 0.9×10^{-2} using over 2000 nodes.

Comment 11

Finally the english is often a bit weak, with some sentences difficult to understand (I have flagged some examples above). Please check thoroughly throughout.
--

Response 11

We greatly appreciate your thorough review of our manuscript. In the revised version, we have carefully proofread the content to enhance the clarity of our writing.

Reference

1. Vandoorne, Kristof, et al. "Experimental demonstration of reservoir computing on a silicon photonics chip." *Nature communications* 5.1 (2014): 3541.
2. Vinckier, Quentin, et al. "High-performance photonic reservoir computer based on a coherently driven passive cavity." *Optica* 2.5 (2015): 438-446.
3. Larger, Laurent, et al. "High-speed photonic reservoir computing using a time-delay-based architecture: Million words per second classification." *Physical Review X* 7.1 (2017): 011015.
4. Rafayelyan, Mushegh, et al. "Large-scale optical reservoir computing for spatiotemporal chaotic systems prediction." *Physical Review X* 10.4 (2020): 041037.
5. Sunada, Satoshi, and Atsushi Uchida. "Photonic neural field on a silicon chip: large-scale, high-speed neuro-inspired computing and sensing." *Optica* 8.11 (2021): 1388-1396.
6. Nakajima, Mitsumasa, Kenji Tanaka, and Toshikazu Hashimoto. "Scalable reservoir computing on coherent linear photonic processor." *Communications Physics* 4.1 (2021): 20.
7. Lupo, Alessandro, et al. "Deep photonic reservoir computer based on frequency multiplexing with fully analog connection between layers." *Optica* 10.11 (2023): 1478-1485.
8. Shen, Yi-Wei, et al. "Deep photonic reservoir computing recurrent network." *Optica* 10.12 (2023): 1745-1751.
9. Antonik, Piotr, et al. "Human action recognition with a large-scale brain-inspired photonic computer." *Nature Machine Intelligence* 1.11 (2019): 530-537.
10. Wang, Hao, et al. "Optical next generation reservoir computing." *arXiv preprint arXiv:2404.07857* (2024).
11. Cox, Nicholas, et al. "Photonic next-generation reservoir computer based on distributed feedback in optical fiber." *arXiv preprint arXiv:2404.07116* (2024).
12. Moon, John, et al. "Temporal data classification and forecasting using a memristor-based reservoir computing system." *Nature Electronics* 2.10 (2019): 480-487.
13. Liu, Keqin, et al. "An optoelectronic synapse based on α -In₂Se₃ with controllable temporal dynamics for multimode and multiscale reservoir computing." *Nature Electronics* 5.11 (2022): 761-773.
14. Liu, Zhuohui, et al. "Interface-type tunable oxygen ion dynamics for physical reservoir computing." *Nature Communications* 14.1 (2023): 7176.
15. Feng, Hanke, et al. "Integrated lithium niobate microwave photonic processing engine." *Nature* 627.8002 (2024): 80-87.
16. Kossel, Marcel A., et al. "8.3 An 8b DAC-based SST TX using metal gate resistors with 1.4 pJ/b efficiency at 112Gb/s PAM-4 and 8-Tap FFE in 7nm CMOS." 2021 IEEE International Solid-State Circuits Conference (ISSCC). Vol. 64. IEEE, 2021
17. Wang, Cheng, et al. "Integrated lithium niobate electro-optic modulators operating at CMOS-compatible voltages." *Nature* 562.7725 (2018): 101-104.
18. Khairi, Ahmad, et al. "A 1.41-pJ/b 224-Gb/s PAM4 6-bit ADC-based SerDes receiver with hybrid AFE capable of supporting long reach channels." *IEEE Journal of Solid-State Circuits* 58.1 (2022): 8-18.
19. Xu, Zhihao, et al. "Large-scale photonic chiplet Taichi empowers 160-TOPS/W artificial general intelligence." *Science* 384.6692 (2024): 202-209.

20. Liang, Xiangpeng, et al. "Rotating neurons for all-analog implementation of cyclic reservoir computing." *Nature communications* 13.1 (2022): 1549.
21. Appeltant, Lennert, et al. "Information processing using a single dynamical node as complex system." *Nature communications* 2.1 (2011): 468.
22. Appeltant, Lennert, et al. "Constructing optimized binary masks for reservoir computing with delay systems." *Scientific reports* 4.1 (2014): 3629.
23. Köster, Felix, et al. "Data-informed reservoir computing for efficient time-series prediction." *Chaos: An Interdisciplinary Journal of Nonlinear Science* 33.7 (2023).
24. Cai, Xinyi, et al. "Scalable photonic reservoir computing based on pulse propagation in parallel passive dispersive links." *Applied Optics* 63.22 (2024): 5785-5791.
25. Wang, Hao, et al. "Optical next generation reservoir computing." *arXiv preprint arXiv:2404.07857* (2024).

Dear Reviewers,

Thank you very much for taking your time to review our manuscript titled "Ultrafast Silicon Photonic Reservoir Computing Engine Delivering Over 200 TOPS" (Manuscript ID: NCOMMS-24-30861A). We would like to express our sincere gratitude for the positive feedback and support from all the reviewers. Meanwhile, we have made additional changes according to Reviewer 3's comments. Detailed point-by-point responses are provided below.

Response To Reviewer 1 (R1)

Comment 1

The authors did a nice job of addressing my concerns and clarifying this work. I appreciated the updated energy analysis and the conceptual figure 1 as well as the clarification of the role the constant power waveguide and the equivalence of the feature vector to a digital NG-RC feature vector. I recommend it for publication.

One typo I noticed: section 4.3 should be data processing, not "date" processing.

Response 1

We sincerely appreciate your support in the publication of our paper. Your thorough review and insightful comments have significantly helped us improve the quality of our manuscript. Additionally, thank you for pointing out the typo, we have now corrected the typo.

Response To Reviewer 2 (R2)

Comment 1

After a careful reading of the authors' responses and the numerous supplements provided, including experimental results, the manuscript proposes content that could be of interest to readers of Nature Communications, also considering the major changes proposed (new title, new figures to present the WDM experiments, etc.). I recommend accepting the manuscript for publication in the journal Nature Communications.

Response 1

Thank you for your positive feedback and support in the publication of our paper. Your thorough review and valuable comments have greatly helped us improve the manuscript quality.

Response To Reviewer 3 (R3)

Comment 1

The authors have significantly improved the manuscript and addressed the referee comments. I believe the manuscript can be published in Nature Communications with a additional changes.

Response 1

Thank you for your positive feedback and support in the publication of our paper. Your thorough review and valuable comments have greatly helped us improve the manuscript quality.

Comment 2

Figure 1 (b) connection between “digital input x ” and “Input: $N+1$ terms”. I would suggest trying to indicate that these connections are all of different lengths, inducing different delays. This is very clear in panel c, but missing in panel b.

Response 2

Thank you for pointing out this issue. Figure 1(b) illustrates the principle of the original next-generation reservoir computing (NG-RC) algorithm implemented in the digital domain. The data points from different time steps need to be reshaped and arranged into the desired sequence for simultaneous processing within the NG-RC framework on a digital processor. As demonstrated in Figure 1(b), this digital implementation of NG-RC processes input information through digital processing rather than relying on physically induced time delays. As a result, we do not indicate different delays in Figure 1 (b). Additionally, we indicate the data points from different time steps as $x_t, x_{t-1}, \dots, x_{t(N-1)}$ in the input layer.

Comment 3

Figure 2 and Section 2.5. In Fig 2, is the Band Pass Filter used in normal operation when a single laser is used? If so why? On the other hand when multiple lasers are used (section 2.5) the BPF may be essential.

I would appreciate having (possibly in the supplementary material) a figure illustrating the experimental setup when multiple wavelengths are used (as it probably differs slightly from Fig 2), and additional explanations on how the system is operated in this case. Are the multiple lasers used at the same time, or sequentially? What BPF is used?

Response 3

Thank you for raising this question. In the single-wavelength operation experimental setup, we used a Band Pass Filter (BPF) to filter out the noise introduced by the Erbium-Doped Fiber Amplifiers (EDFA). We use the DiCon’s Manually Tunable Bandpass Filter (Tuning Range: 1535 – 1565nm, 0.5 dB Bandwidth: 0.8 nm) in all experiments.

According to your suggestion, we have added a new figure (Supplementary Fig.3) in the

supplementary material to illustrate the experimental setup for multi-wavelength operation, along with additional explanations on how the system functions in this configuration. In our experiment, multiple lasers are used sequentially. However, through wavelength-division-multiplexing, it is straightforward to implement multiple lasers simultaneously using the setup shown in Supplementary Fig.3.

Comment 4

Energy Consumption and Supplementary Note 4. I do not understand how the authors take into account the energy consumption in the system they use. For instance the Fast Oscilloscope consumes approximately 1kW (not 5.1 W). I believe the authors should include a comparison of: 1) energy consumption in the system they implemented (including consumption of AWG, oscilloscope, current sources, laser driver, etc.); 2) energy consumption of a realistic system with accessible components.

Similarly, does Fig S3 refer to the real experimental system, or to a future system? Please clarify.

Response 4

Thank you for pointing out the issue. In the revised supplementary information, we have provided more detailed information about how the energy consumption is estimated.

Our all-optical engine system consists of a laser, an erbium-doped fiber amplifier (EDFA), a 256 GSamples/s digital-to-analog converter (DAC), an on-chip modulator, ninety metal heaters, and an on-chip balanced photodetector paired with a 256 GSamples/s analog-to-digital converter (ADC). Their power consumption is estimated as follows:

1. The laser used in our experiment is a PPCL500 with a power consumption of approximately 1W.
2. The energy consumption of EDFA is estimated as $\frac{\lambda_s}{\lambda_p}(P_{out}^s - P_{in}^s)/\eta$ [1]. Here, the signal wavelength is $\lambda_s=1,550$ nm, the pump wavelength is $\lambda_p=1,480$ nm, the output power is $P_{out}^s \approx 20$ dBm, the input power is $P_{in}^s \approx 0$ dBm, and the wall-plug efficiency is $\eta \approx 0.3$. Based on these parameters, the energy consumption of EDFA is approximately 0.35 W.
3. To estimate the power consumption for the optical signal generation and detection, we assume that it can be realized by an optical transceiver module, rather than the arbitrary waveform generator (AWG) and real-time oscilloscope (RTO) equipment we used in the lab. The DAC-based transmitter in [2] achieves an energy efficiency of 2.8 pJ per Baud. The modulator in [3] achieves an energy efficiency of 42 fJ per Baud. In addition, the energy consumption of the modulator bias is approximately 25 mW. Therefore, the combined energy consumption of the DAC (operating at a sampling rate of 256 GSamples/s) and the modulator is around 0.75W.
4. Based on our experimental measurements, the power consumption of the metal heater phase shifter for achieving a 2π phase shift is approximately 50 mW. We assume that each metal heater in our system achieves an average π phase shift. Thus, the total energy consumption of

the ninety metal heaters is approximately 2.25 W.

5. The ADC-based receiver in [4] achieves an energy efficiency of 2.82 pJ per Baud. The PD in [5] achieves an energy efficiency in the range of a few fJ/bit. Therefore, the total energy consumption of the ADC (operating at a sampling rate of 256 GSamples/s) and the PD is approximately 0.75 W.

To sum up, the total energy consumption of the all-optical engine is about 5.1 W.

The reviewer is correct that the AWG and RTO for lab use can be very power-consuming. But we think most of their power consumption is not necessary for our application, as they not only incorporate DACs or ADCs but also integrate additional complex functions that we don't need such as real-time waveform processing, display, spectrum analysis, triggering, demodulation, signal storage, and mathematical operations. Therefore, when estimating power consumption, we use the energy consumption of DAC-based transmitters and ADC-based receivers as a reference.

Thank you for your reminder regarding the issue of the original Fig. S3. The original Fig. S3 refers to the future all-optical engine. We have updated the caption of this figure (Fig. S4 in updated supplementary information) to clarify that it represents the all-optical engine.

Comment 5

The supplementary material contains a new table that is illustrated by figures taken from a dozen publications. The authors need to check that they have authorization to publish these figures.
--

Response 5

Thank you for your reminder regarding this issue. We have removed the figures from Supplementary Table S1.

Reference

1. Feng, Hanke, et al. "Integrated lithium niobate microwave photonic processing engine." *Nature* 627.8002 (2024): 80-87.
2. M. A. Kossel, V. Khatri, M. Braendli, et al., "8.3 an 8b dac-based sst tx using metal gate resistors with 1.4 pj/b efficiency at 112gb/s pam-4 and 8-tap ffe in 7nm cmos," in 2021 IEEE International Solid-State Circuits Conference (ISSCC), vol. 64 (IEEE, 2021), pp. 130–132.
3. Wang, Cheng, et al. "Integrated lithium niobate electro-optic modulators operating at CMOS-compatible voltages." *Nature* 562.7725 (2018): 101-104.
4. Khairi, Ahmad, et al. "A 1.41-pJ/b 224-Gb/s PAM4 6-bit ADC-based SerDes receiver with hybrid AFE capable of supporting long reach channels." *IEEE Journal of Solid-State Circuits* 58.1 (2022): 8-18.
5. Benedikovic, Daniel, et al. "Silicon-germanium avalanche receivers with fJ/bit energy consumption." *IEEE Journal of Selected Topics in Quantum Electronics* 28.2: Optical Detectors (2021): 1-8.